# SERCA2 phosphorylation at serine 663 is a key regulator of Ca²⁺ homeostasis in heart diseases

Fabrice Gonnot[1], Laura Boulogne [1,7], Camille Brun [1,7], Maya Dia[1], Yves Gouriou[1], Gabriel Bidaux[1], Christophe Chouabe [1], Claire Crola Da Silva[1], Sylvie Ducreux [1], Bruno Pillot [1], Andrea Kaczmarczyk[1], Christelle Leon [1], Stephanie Chanon[2], Coralie Perret[1], Franck Sciandra[1], Tanushri Dargar [3], Vincent Gache[3], Fadi Farhat[1,4,5], Laurent Sebbag[1,4,6], Thomas Bochaton[1,4], Helene Thibault[1,4], Michel Ovize[1,4], Melanie Paillard [1] & Ludovic Gomez [1] ✉

Despite advances in cardioprotection, new therapeutic strategies capable of preventing ischemia-reperfusion injury of patients are still needed. Here, we discover that sarcoplasmic/endoplasmic reticulum Ca²⁺ ATPase (SERCA2) phosphorylation at serine 663 is a clinical and pathophysiological event of cardiac function. Indeed, the phosphorylation level of SERCA2 at serine 663 is increased in ischemic hearts of patients and mouse. Analyses on different human cell lines indicate that preventing serine 663 phosphorylation significantly increases SERCA2 activity and protects against cell death, by counteracting cytosolic and mitochondrial Ca²⁺ overload. By identifying the phosphorylation level of SERCA2 at serine 663 as an essential regulator of SERCA2 activity, Ca²⁺ homeostasis and infarct size, these data contribute to a more comprehensive understanding of the excitation/contraction coupling of cardiomyocytes and establish the pathophysiological role and the therapeutic potential of SERCA2 modulation in acute myocardial infarction, based on the hotspot phosphorylation level of SERCA2 at serine 663 residue.

Ischemic heart disease (IHD) is one of the leading causes of death and disability worldwide[1]. Prevalence of acute myocardial infarction (AMI) as the first manifestation of IHD is approximatively 50%. It manifests clinically as arrhythmias, myocyte death and contractile dysfunction[2,3]. Patients surviving AMI are susceptible to secondary events: recurrent reinfarction, arrhythmias, hypoperfusion, sudden cardiac death and above all, heart failure (HF)[4]. Over the past three decades, despite cardioprotection strategies have been developed in animal models[5,6], the translation to human failed to achieve better clinical outcomes in

patients[7]. Therefore, novel therapeutic strategies that can prevent acute myocardial ischemia-reperfusion (I/R) injury and reduce secondary events, are still needed.

Calcium ion (Ca²⁺) is an ubiquitous signal regulating several cellular functions, including survival and death[8]. Growing evidence suggests that an increased cytosolic free Ca²⁺ overload is one of the major contributors of myocardial I/R injury[9]. Ca²⁺ handling in the cardiac muscle is coordinated by a set of proteins that include Na⁺/Ca²⁺ exchangers, L-type Ca²⁺ channels, ryanodine receptors and sarco/

[1]Laboratoire CarMeN - IRIS Team, INSERM, INRA, Université Claude Bernard Lyon-1, INSA-Lyon, Univ-Lyon, 69500 Bron, France. [2]Laboratoire CarMeN, INSERM, INRA, Université Claude Bernard Lyon-1, INSA-Lyon, Univ-Lyon, Functional Lipidomic Plateform, Lyon, France. [3]Institut NeuroMyoGène INMG-PNMG, CNRS UMR5261, INSERM U1315, Université Claude Bernard Lyon 1, Lyon, France. [4]Hôpital Louis Pradel, Hospices Civils de Lyon, 59 boulevard Pinel, F-69500 Bron, France. [5]Cardiac Surgery Department, Hospices Civils de Lyon, Hôpital Louis Pradel, 69500 Bron, France. [6]Heart Failure and Transplant Department, Hospices Civils de Lyon, Hôpital Louis Pradel, 69500 Bron, France. [7]These authors contributed equally: Laura Boulogne, Camille Brun. ✉e-mail: Ludovic.gomez@univ-lyon1.fr

endoplasmic reticulum $Ca^{2+}$-ATPases (SERCA2)[10]. In failing hearts, the deficiency of sarco/endoplasmic reticulum (SR/ER) $Ca^{2+}$ uptake in cardiomyocytes is mainly explained by a decrease in the expression and activity of the SERCA2 pump[11]. Therefore, SERCA2 appears as a crucial therapeutic avenue to be targeted as soon as the reperfusion phase of AMI in order to prevent the evolution towards contractile dysfunction and HF. Although SERCA2 is one of the most promising targets for the treatment of contractile dysfunction following I/R, recent gene therapy trials failed to show a benefit of SERCA2 over-expression in patients[12,13], indicating that the development of new modalities to enhance SERCA2 activity to improve patient outcome is needed.

It has been shown that the main regulators of SERCA2 proteins are phospholamban (PLN) in the ventricles, and sarcolipin in the atria[14]. Recent therapies have been proposed by targeting PLN modulation of SERCA2 in arrhythmia and HF[15,16]. However, recent findings suggest that SERCA2 activation can occur without dissociation of the SERCA2/PLN complex[17,18], suggesting that other phenomenon could modulate SERCA2 activity. Among them, post-translational modifications of SERCA2[19–24] have been proposed to regulate SERCA2 activity including phosphorylation[25,26]. In this context, recent studies have reported that the glycogen synthase kinase 3 beta (GSK3β) was an important regulator of cellular function[27], and a key enzyme in the myocardial response to I/R injury[28–30]. Moreover, we and others demonstrated that GSK3β regulates $Ca^{2+}$ homeostasis and diastolic function in the heart[31]. Noteworthily, the cardioprotection mediated by GSK3β inhibition limits $Ca^{2+}$ overload at reperfusion[32]. Here, taking advantage of the physiological characteristics of the GSK3β kinase in the cardioprotection field, and combining in silico analysis with in vitro and in vivo genetic approaches, we report serine 663 phosphorylation as a physiological and clinical event in the SERCA2 regulation in heart, and that its limitation improves cardiomyocyte function and prevents cell death and subsequent myocardial infarct size.

## Results

### SERCA2 is phosphorylated at serine 663 in both mouse and human hearts

The cardiac isoform 2 of the sarco/endoplasmic reticulum $Ca^{2+}$ ATPase (SERCA2) plays a major role in controlling the excitation/contraction coupling (ECC). We recently identified a novel role for GSK3β as an important regulator of $Ca^{2+}$ transfer between SR/ER and mitochondria in heart[32]. To decipher a potential role for GSK3β in modulating SERCA2 activity, we first evaluated the effect of the pharmacological inhibition of GSK3β on SERCA2 activity, by monitoring the ER $Ca^{2+}$ pumping after $Ca^{2+}$ depletion (Fig. 1A). We found that cells treated with SB216763 or TDZD8 exhibited a twofold increase in the rate of ER $Ca^{2+}$ refilling as compared to control cells (Fig. 1B), suggesting a causal link between GSK3β activity and the SERCA2 pumping activity. To test this hypothesis, because GSK3β is a proline-directed serine-threonine kinase, we next performed a SERCA2 in silico analysis for putative GSK3β phosphorylation sites[33,34] (Fig. 1C). Our analysis revealed 11 consensus phosphorylation sites of SERCA2 by GSK3β (Fig. 1D), including 9 sites in the cytoplasmic domain (Supplementary Fig. 1A).

To refine our subsequent analysis on these potential phosphorylation sites, we next questioned if these sites could be a physiological event of a cardiovascular disease. Combining an in vivo mouse model of I/R with a mass spectrometry approach, we revealed that SERCA2 is phosphorylated at serine 663 (S663) in the mouse heart in basal (Fig. 1E), and that regional cardiac I/R significantly increased by 21% the phosphorylation level of SERCA2 at the S663 residue (Fig. 1E). To translate these fundamental results to the clinical level, the experiment described above was repeated on human heart samples. In line with our previous results, we found that a pool of SERCA2 is phosphorylated at S663 in healthy human (non-failing) hearts (Fig. 1F), and that the human failing heart exhibited a significantly fourfold increase in

SERCA2 phosphorylation at S663. These results demonstrate that the S663 phosphorylation of SERCA2 is increased by ischemia-reperfusion in a preclinical model, an increase also present in the human failing heart. Phosphorylation of SERCA2 on S378 and S555 was also identified in the mouse model; however, only the S378 phosphorylation was significantly increased in the ischemic mouse heart averaging $22.43 \pm 0.3\%$ vs $6.65 \pm 0.09\%$ in control group. These sites were not further investigated as they were not identified as in silico putative GSK3β phosphorylation sites, and were not identified by the MS approach on human samples.

Interestingly, S663 is positioned on the cytosolic side of SERCA2 and easily accessible to kinases (Fig. 1G), notably GSK3β. To determine a potential involvement of GSK3β on SERCA2 phosphorylation, we used three different strategies to assess the probability of the physical interaction between GSK3β and SERCA2 and the effect of simulated ischemia-reperfusion. First, immunoprecipitation assays demonstrated that GSK3β formed a complex with SERCA2 in both basal condition and after hypoxia-reoxygenation (H/R) in HEK cells (Fig. 1H). Second, our designed fluorescence resonance energy transfer (FRET) system validated the intermolecular interaction of GSK3β with SERCA2 in single live HEK cells (Fig. 1I, left; Supplementary Fig. 1B-D), with a FRET efficiency of 3.11% in normoxic condition (Fig. 1I right), which exhibited a significant threefold increase following H/R (Fig. 1I, right). Third, a proximity ligation assay demonstrated an increased proximity between endogenous GSK3β and SERCA2 proteins in isolated mouse cardiomyocytes after H/R (Fig. 1J). Accordingly, these results provide valuable insight regarding a key regulatory mechanism of SERCA2, based on phosphoserine-mediated recruitment of GSK3β to SERCA2 in the heart during an ischemic stress.

### Phosphorylation state of SERCA2 at S663 modulates cellular $Ca^{2+}$ transfer profiles

To decipher the role of S663 phosphorylation as a regulator of SERCA2 activity, a CRISPR/Cas9-mediated genome editing strategy allowed us to generate a human cell line expressing a phosphoresistant SERCA2 mutant (SERCA2[S663A]). The evaluation of SERCA2 activity revealed that, similarly to the GSK3β inhibitors (Fig. 1A, B), the velocity of ER $Ca^{2+}$ refilling was significantly increased by 1.8-fold in the SERCA2[S663A] cells as compared to control (WT) (Fig. 2 A, B). Furthermore, the ER $Ca^{2+}$ content was significantly higher in the SERCA2[S663A] cells (Fig. 2C), while both cytosolic and mitochondrial $Ca^{2+}$ contents were significantly reduced in basal condition (Fig. 2D, E). Interestingly, the addition of a GSK3β inhibitor showed no synergistic effect on the $Ca^{2+}$ refilling rate in SERCA2[S663A] cells (Fig. 2A, B). This indicates that either SERCA2[S663A] mutant dominates GSK3β-dependent inhibition of SERCA2 activity or that the GSK3β-dependent inhibition relies on S663 residue to repress SERCA2 activity.

To rule out a possible adaptive phenotype on the $Ca^{2+}$ homeostasis of the HEK mutant cells, we next challenged the phosphoresistant cells with an acute ER $Ca^{2+}$ release and measured both cytosolic and mitochondrial $Ca^{2+}$ content. Following IP3R-induced ER $Ca^{2+}$ release, cells expressing the SERCA2[S663A] mutant exhibited a significantly 1.4-fold decreased cytosolic $Ca^{2+}$ peak amplitude (Supplementary Fig. 2A, B) and a 1.8-fold decreased mitochondrial $Ca^{2+}$ peak amplitude (Supplementary Fig. 2C, D) as compared to WT cells. These results reinforce the notion that SERCA2 activity is increased in SERCA2[S663A] cells, allowing the rapid uptake of ER $Ca^{2+}$ release into the ER, therefore limiting the transit of $Ca^{2+}$ in the cytosol or other organelles such as mitochondria.

One may question whether SERCA2[S663A] mutant may also impact the induced ER $Ca^{2+}$ release, which may explain the decrease in both cytosolic and mitochondrial $Ca^{2+}$ contents. To answer this question, we repeated $Ca^{2+}$ release experiments in presence of thapsigargin, an irreversible SERCA2 pump inhibitor. In these conditions, the IP3R-induced ER $Ca^{2+}$ release caused a similar $Ca^{2+}$ increase in both cytosolic

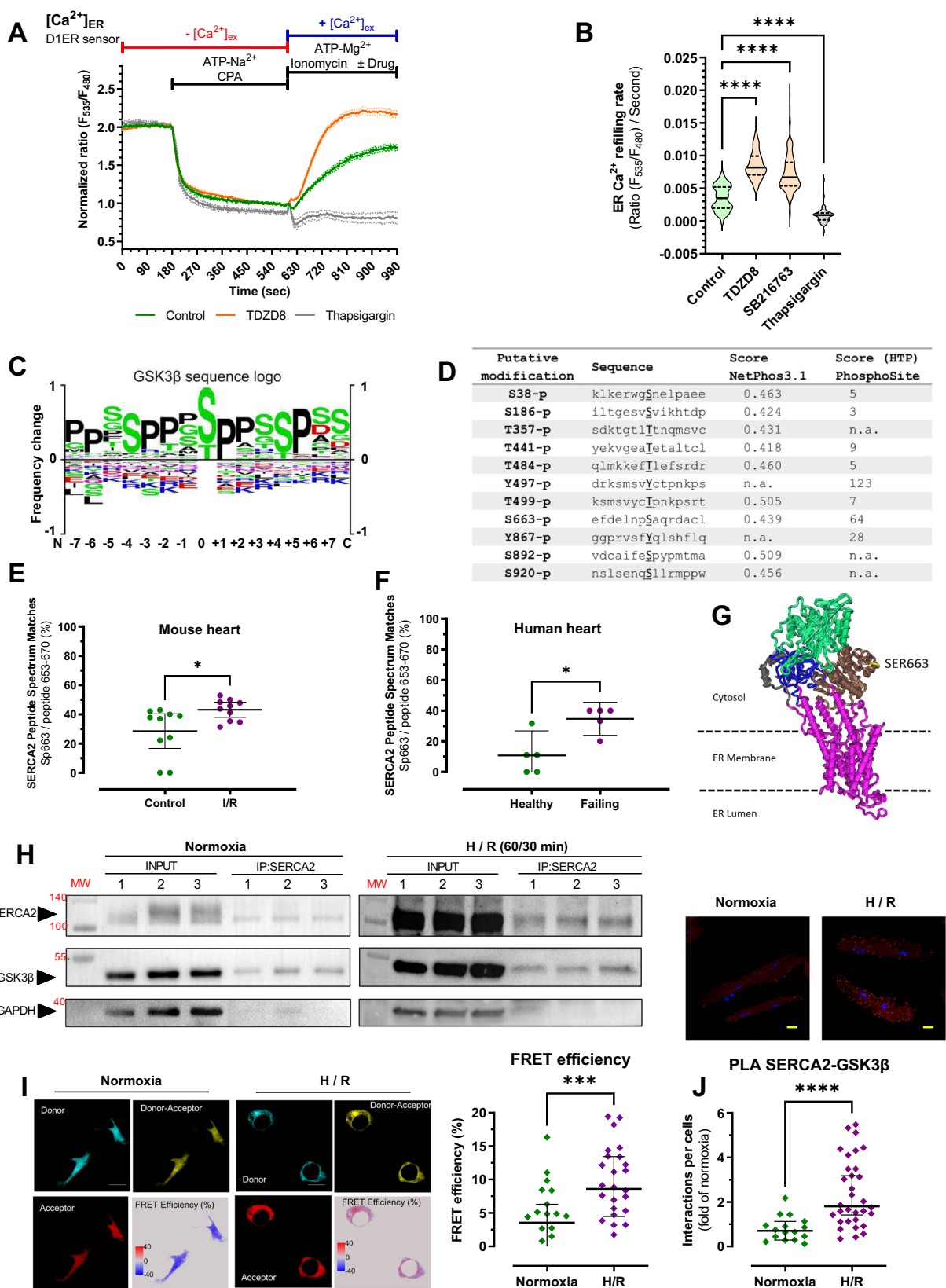

and mitochondrial compartments in all groups (Supplementary Fig. 2E–H), indicating that the same quantity of $Ca^{2+}$ was released from the ER in all groups, and confirming that the differences observed in Supplementary Fig. 1A–D were linked to the increase of the ER $Ca^{2+}$ uptake rather than to a decrease of ER $Ca^{2+}$ release or other possible adaptive modifications in the SERCA2[S663A] cells.

To validate the ubiquitous role of SERCA2 phosphorylation on S663, we performed a similar experiment in human induced pluripotent stem-cell-derived cardiomyocytes (hiPSC-CM) since SERCA2 plays a crucial role in cardiomyocytes[35]. hiPSC-CM were infected by either a WT SERCA2, a phosphoresistant SERCA2 mutant (SERCA2[S663A]) or a phosphomimetic SERCA2 mutant (SERCA2[S663E])

**Fig. 1 | GSK3β-dependent phosphorylation of SERCA2 at serine 663 in human and mouse hearts. A** Representative curves of normalized $[Ca^{2+}]_{ER}$ ratio signals over time measured with the genetically encoded $Ca^{2+}$ probe D1ER in HEK293-T cells in $Ca^{2+}$ free and 1 mM EGTA-containing buffer for three minutes before depleting the ER $Ca^{2+}$ store by adding 100 μM cyclopiazonic acid (CPA) and 500 μM ATP-$Na^{2+}$ for seven minutes and prior to ER $Ca^{2+}$ refilling by switching for six minutes and a half to 2 mM $Ca^{2+}$, 500 μM ATP-$Mg^{2+}$ and 1 μM ionomycin-containing buffer (green curve), or supplemented with 10 μM TDZD8 (orange curve), or 1 μM thapsigargin (gray curve) to compare SERCA activity. Ratio signals were normalized as following: $[1+ (R − R_{t=600s})/(R_{t=180s.} − R_{t=600s})]$. **B** Violin plot of ER $Ca^{2+}$ refilling slope over two minutes from time 630 s of control ($n = 199$, on 11 experimental days), TDZD8 ($n = 210$, on 7 experimental days), thapsigargin ($n = 42$, on 2 experimental days) and 1 μM SB216763 (N = 212 cells, on 6 experimental days). Median with a 95% confidence interval is shown and one-way ANOVA followed by Tukey's multiple comparisons test was used to assess significance versus Control condition (****$p < 0.0001$). **C** Motif of the GSK3β substrate sequence (from PhosphoSitePlus database). **D** Listing of in silico analysis of putative GSK3β target sites on SERCA2 determined by NetPhos3.1 and PhosphoSitePlus public databases. **E** Mass spectrometry analysis of SERCA2 phosphorylation at serine 663 residue (Sp663) in left ventricle of C57BL/6 mice in basal (Control; $n = 10$) and after 60 min ischemia followed by 30 minutes reoxygenation (IR; $n = 10$). Median with interquartile range is shown and two-tailed Mann–Whitney test was used to assess significance (*$p = 0.027$). **F** Mass spectrometry analysis of SERCA2 phosphorylation at serine 663

in healthy patients (Non-failing; $n = 5$) and patient with end-stage heart failure (Failing; $n = 5$ regions). Median with interquartile range is shown and two-tailed Mann–Whitney test was used to assess significance (*$p = 0.0159$). **G** 3D structure of SERCA2, with α-helix and β-sheet in different SERCA2 domains. Visualization of the accessibility of the cytosolic phosphorylation site at serine 663 (SER663, yellow) at the periphery of the phosphorylation domain of SERCA2 (brown). Figure adapted from[71]. **H** Interaction of GSK3β with SERCA2 determined by co-immunoprecipitation on lysates from HEK293-T cells (basal ($n = 3$) and after 1 h hypoxia followed by 30 min reoxygenation ($n = 3$)). GAPDH was used as a negative control. **I** Representative images of FRET between SERCA2-mTurquoise2 donor and GSK3β-sYFP2 acceptor, on living HEK293-T cells, in normoxic condition or after 60 min hypoxia followed by 30 minutes reoxygenation (H/R). Scale bar represents 10μm (left). Dot plot of the FRET efficiency in normoxia ($n = 15$) and after H/R ($n = 24$, ***$p = 0.0009$), on four independent experiments (right). Median with interquartile range is shown and two-tailed Mann–Whitney test was used to assess significance. **J** Representative confocal microscopy images of the in situ SERCA2-GSK3β proximity depicted as red dots on adult cardiomyocytes, in normoxic or after 45 minutes hypoxia followed by 120 min reoxygenation (H/R). Nuclei appear in blue (top). Quantification of the interactions per cell of normoxia ($N = 16$) and H/R ($N = 32$) groups, presented as a fold of Normoxia, on three independent experiments. Scale bar represents 10 μm (bottom). Median with interquartile range is shown and two-tailed Mann–Whitney test was used to assess significance (****$p < 0.0001$).

(Fig. 2F). Similarly to what was observed in the HEK cells, the SERCA2[S663A] hiPSC-CM displayed significantly increased ER $Ca^{2+}$ refilling rate and ER $Ca^{2+}$ content versus the SERCA2[WT] cells (Fig. 2G, H). On the other hand, the SERCA2[S663E] hiPSC-CM showed no difference in their ER $Ca^{2+}$ refilling rate compared to the SERCA2[WT] cells, but a significant decrease in the ER $Ca^{2+}$ content compared to both SERCA2[WT] and SERCA2[S663A] hiPSC-CM (Fig. 2G, H).

Taken together, these results demonstrate that specific inhibition of SERCA2 phosphorylation at S663 residue increases SERCA2 pump activity in several human cell types, notably hiPSC-CM.

### Phosphorylation state of SERCA2 at S663 controls cell death after hypoxia-reoxygenation (H/R)

Failing hearts after AMI are characterized by $Ca^{2+}$ deregulation, cell death activation and deleterious kinase expression. We hypothesized that the increase in SERCA2 activity (S663-dependent) could detoxify cytosolic $Ca^{2+}$ overload, prevent mitochondrial $Ca^{2+}$ overload and therefore protect against cell death at reperfusion. To test our hypothesis, we first performed in vitro hypoxia-reoxygenation on HEK cells. Our results revealed that SERCA2[S663A] mutant conferred around 50% protection as compared to control (WT) (Fig. 3A). Evaluation of $Ca^{2+}$ kinetics during the first hours of reoxygenation indicated that the SERCA2[S663A] cells refilled ER $Ca^{2+}$ stores more efficiently than control cells (WT) (Fig. 3B). Consequently, lower cytosolic and mitochondrial $Ca^{2+}$ levels were measured during reoxygenation in the SERCA2[S663A] mutant (Fig. 3C, D). These results imply that SERCA2 phosphorylation at S663 participates to both cytosolic and mitochondrial $Ca^{2+}$ overload, which ultimately leads to cell death.

To reinforce our results, we engineered a mouse embryonic fibroblast (MEF) cell lines with a *Serca2-null* background, to express a phosphomimetic (rS663E) form of SERCA2 at serine 663. In contrast to SERCA2-phosphoresistant mutant, the constitutive phosphorylation of SERCA2 at S663 is associated with a cellular $Ca^{2+}$ overload and an exacerbated cell death (Fig. 3E–G).

In parallel, to validate the translational value of our protection strategy, we infected hiPSC-CM cells with AAV9-phosphoresistant mutant. Similarly to what was observed in both HEK and MEF cells, SERCA2[S663A] hiPSC-CM cells displayed a significant decreased cell death after H/R stress as compared to SERCA2[WT] control group (Fig. 3H). Altogether, our results provide an essential regulatory mechanism of $Ca^{2+}$ homeostasis and cell death during H/R, based on

the phosphorylation state of SERCA2 at S663 in several human cell types, notably hiPSC-CM.

### Precluding S663-SERCA2 phosphorylation ameliorates SR/ER $Ca^{2+}$ uptake and excitation-contraction coupling in mouse cardiomyocytes

To examine the physiological function of SERCA2 phosphorylation status at serine 663 in isolated adult cardiomyocytes, we generated three adeno-associated viruses (AAV) containing either the wild-type (rWT), the phosphoresistant (rS663A) or the phosphomimetic (rS663E)-SERCA2 mutant, which were injected in cardiomyocyte-specific inducible SERCA2-knock down (KD) mice[36] (Supplementary Fig. 3A). All analyzed mice presented a cardiomyocyte-specific excision of the endogenous *Serca2* (Supplementary Fig. 3B, averaging 84%) and a SERCA2 mutant rescue, as displayed by immunoblotting (Supplementary Fig. 3C). Validation of the AAV9 expression in the whole heart after jugular injection was further confirmed using the fluorescent AAV9-D4ER and the AAV9-LacZ (Supplementary Fig. 3D). After having verified that the phosphorylation level of phospholamban at both serine 16 and threonine 17 was not modified in our basal mutant cardiomyocytes (Supplementary Fig. 3E), we next evaluated the ECC properties of the different SERCA2-rescued cardiomyocytes using $Ca^{2+}$ imaging and electrophysiological measurements.

Monitoring cytosolic $Ca^{2+}$ transients under field stimulation at 1 Hz (Fig. 4A) demonstrated that the $Ca^{2+}$ transient amplitude was significantly increased in rS663A as compared to rWT and rS663E cardiomyocytes (Fig. 4B), suggesting that the SR/ER $Ca^{2+}$ stores were larger in cardiomyocytes expressing the phosphoresistant mutant. To note, no difference in the time-to-peak was observed between mutant groups (Supplementary Fig. 3F), suggesting no change in RyR-induced $Ca^{2+}$ release. Consecutively to electrical stimulations, we next evaluated the RyR-dependent reticular $Ca^{2+}$ pool using caffeine stimulus (Fig. 4A, arrow). As expected, the maximal caffeine-peak amplitude of rS663E mutant was significantly reduced as compared to rS663A and rWT groups (Fig. 4C). However, we were surprised to visualize a similar answer in rS663A and rWT (Fig. 4C). This apparent discrepancy can be explained by a faster $Ca^{2+}$ SERCA2-dependent pumping back into the SR/ER in the rS663A cardiomyocytes than in rWT and rS663E cells, confirmed by the higher rate of exponential decay in these cells (Fig. 4D). Accordingly, rS663A cardiomyocytes displayed a lower cytosolic $Ca^{2+}$ level at baseline (Fig. 4E), supporting an enhanced SR $Ca^{2+}$ pumping.

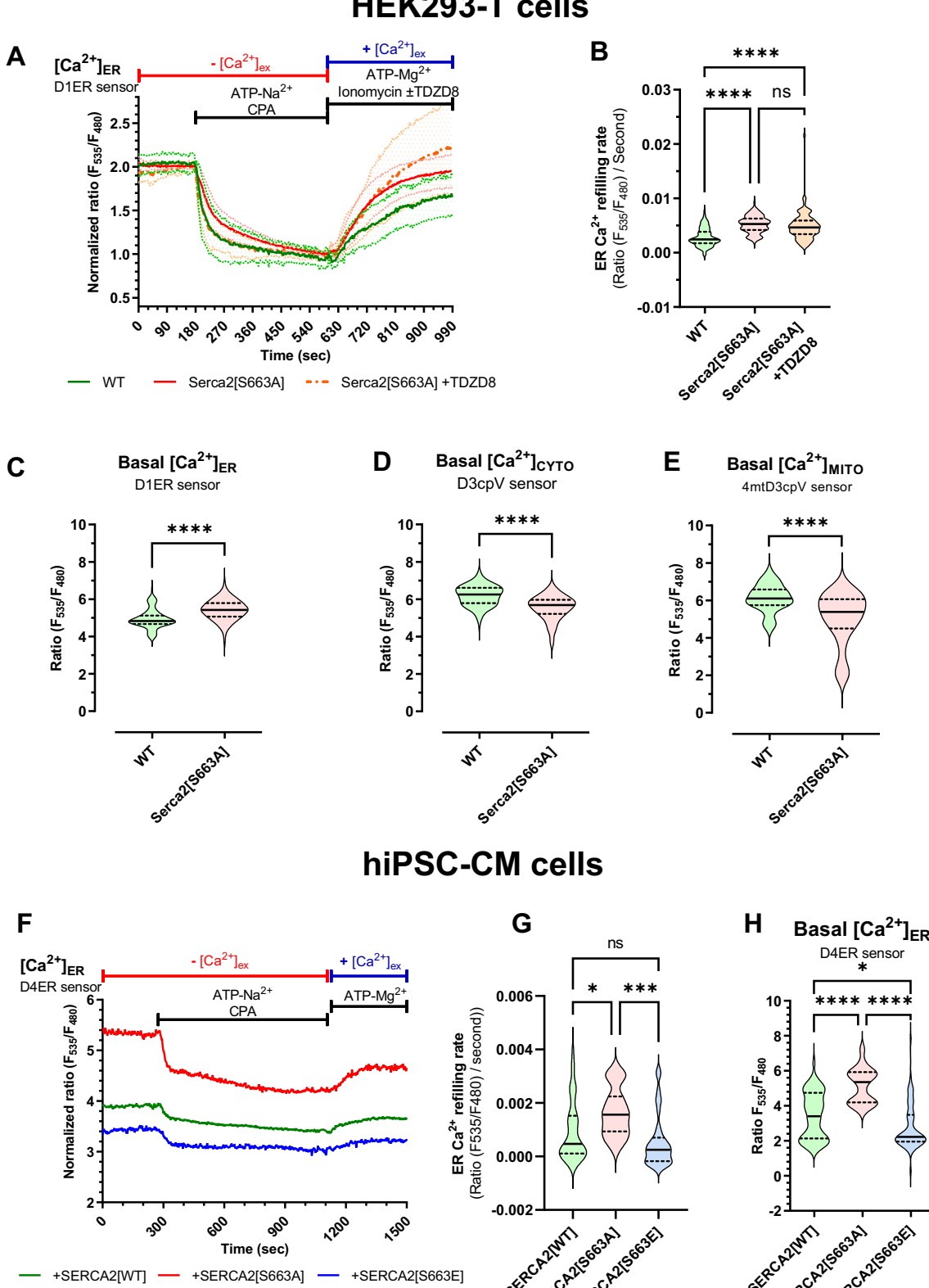

**HEK293-T cells**

**hiPSC-CM cells**

We next performed electrophysiological recordings in mouse cardiomyocytes from the three mutants in order to characterize the two major $Ca^{2+}$ fluxes across the sarcolemma involved in ECC: (i) the L-type $Ca^{2+}$ current ($I_{Ca,L}$) and (ii) the $Na^{+}/Ca^{2+}$ exchange current ($I_{NCX}$). No significant difference was reported between the capacitances and $I_{Ca,L}$ densities of rWT, rS663A, and rS663E mutant cardiomyocytes

(Fig. 4F). Likewise, $I_{NCX}$ densities measured as the $Li^{+}$-sensitive slow tail current 20 ms after the onset of repolarization at −80 mV (Fig. 4G) or as the integrated form of this inward current (Fig. 4H) were similar in the 3 subgroups. To note, when compared to cardiomyocytes isolated from non-infected control (Ctr) mice, their capacitance and $I_{Ca,L}$ density were comparable to those of mutant cells (Fig. 4F). However, the

**Fig. 2 | Phosphorylation of SERCA2 at serine 663 regulates SERCA2 activity.**
**A** Representative curves of normalized $[Ca^{2+}]_{ER}$ ratio signals over time measured with the genetically encoded $Ca^{2+}$ probe D1ER in HEK293-T wild-type (green curve), Serca2[S663A] phosphoresistant mutant (red curve) and Serca2[S663A] phosphoresistant mutant supplemented with 10 μM TDZD8 for ER $Ca^{2+}$ refilling (dashed orange curve). Ratio signals were normalized as following: $[1 + (R − R_{t=600s})/(R_{t=180sec.} − R_{t=600s})]$. **B** Violin plot of the ER $Ca^{2+}$ refilling slope over two minutes from time 630 s of WT ($N = 108$ cells on 4 experimental days), Serca2[S663A] ($N = 108$ cells on 9 experimental days) and Serca2[S663A] + TDZD8 ($N = 100$ cells on 4 experimental days) respectively. Median with a 95% confidence interval is shown and one-way ANOVA followed by Tukey's multiple comparisons test was used to assess significance (****$p < 0.0001$) (ns $p = 0.3953$, ****$p < 0.0001$). **C** Violin plot of basal $[Ca^{2+}]_{ER}$ (endoplasmic reticulum) ratio measured with D1ER probe in HEK293-T wild-type (WT) ($N = 117$) and Serca2[S663A] phosphoresistant mutant cells ($N = 108$). Median with distribution is shown and two-tailed unpaired $t$ test with Welch's correction was used to assess significance (****$p < 0.0001$). **D** Violin plot of basal $[Ca^{2+}]_{CYTO}$ (cytosol) ratio measured with D3cpV probe in HEK293-T wild-type (WT) ($N = 214$) and Serca2[S663A] phosphoresistant mutant cells ($N = 124$). Median with distribution is shown and two-tailed unpaired $t$ test with Welch's correction was used to assess significance (****$p < 0.0001$). **E** Violin plot of basal $[Ca^{2+}]_{MITO}$

(mitochondria) ratio measured with 4mtD3cpV probe, in HEK293-T wild-type (WT) ($N = 219$) and Serca2[S663A] phosphoresistant mutant cells ($N = 197$). Median with distribution is shown and two-tailed unpaired $t$ test with Welch's correction was used to assess significance (****$p < 0.0001$). **F** Representative curves of normalized $[Ca^{2+}]_{ER}$ ratio signals over time measured with the genetically encoded $Ca^{2+}$ probe D4ER in human induced pluripotent stem-cell-derived cardiomyocytes (hiPSC-CM) infected (6 days, MOI 100 000) with the AAV9-Serca[WT], wild-type (green curve), the AAV9-Serca2[S663A] phosphoresistant mutant (red curve) and the AAV9-Serca2[S663E] phosphomimetic mutant (blue curve). **G** Violin plot of ER $Ca^{2+}$ refilling slope over four minutes from time 1119 s in AAV9-Serca[WT] ($N = 44$ cells on 4 experimental days), AAV9-Serca2[S663A] ($N = 19$ cells on 4 experimental days) and AAV9-Serca2[S663E] ($N = 35$ cells on 4 experimental days) infected hiPSC-CM cells. Median with a 95% confidence interval is shown and one-way ANOVA followed by Tukey's multiple comparisons test was used to assess significance (ns $p = 0.2212$, *$p = 0.0296$, ***$p = 0.0007$). **H** Violin plot of basal $[Ca^{2+}]_{ER}$ (endoplasmic reticulum) ratio signal measured with D4ER probe (6 days, MOI 100 000) in hiPSC-CM infected with AAV9-Serca2[WT] ($N = 57$ on 4 experimental days), -Serca2[S663A] ($N = 19$ on 4 experimental days) or -Serca2[S663E] ($N = 64$ on 4 experimental days). Median with a 95% confidence interval is shown and one-way ANOVA followed by Tukey's multiple comparisons test was used to assess significance (*$p = 0.0115$, ****$p < 0.0001$).

$I_{NCX}$ density of the Ctr cardiomyocytes was higher than all 3 mutants (Fig. 4G, H), without being related to a difference in NCX protein expression (Supplementary Fig. 3G).

In the next protocol (Fig. 4I–K), NCX currents, reflecting the $Ca^{2+}$ cytosolic content, were used to test the competition between sarcolemmal $Ca^{2+}$ efflux (primarily through the NCX) and $Ca^{2+}$ uptake into SR/ER (through SERCA2) that governs relaxation[37]. Cardiomyocytes were thus challenged with an episode of intracellular $Ca^{2+}$ overload achieved by triggering the $Ca^{2+}$-induced $Ca^{2+}$ release (CICR) after lithium blockade of NCX currents. In the presence of external $Na^+$, each pacing protocol (Fig. 4J, P1–P3) triggered a brief tonic cell contraction. Cardiomyocytes gradually lost their ability to contract normally in the presence of external $Li^+$ (Fig. 4J, P4–P6) because of the cytosolic $Ca^{2+}$ increase caused by the cessation of the NCX-dependent $Ca^{2+}$ efflux across the sarcolemma. On return to external $Na^+$, the slow tail inward current recorded on repolarization was almost tripled in every cardiac cell type (Fig. 4J: $P_7$ trace on return to external $Na^+$ versus $P_3$ trace before external $Li^+$), showing that NCX exchangers worked properly. However, the return to its value before external $Li^+$ addition was faster in any of the 3 mutants than in non-infected cells, and even more markedly in the rS663A mutant cardiomyocytes. When only considering the mutants, the integrated NCX current measured after return to external $Na^+$ was significantly smaller in the phosphoresistant rescue as compared to rWT (from $P_8$ to $P_{14}$) and phosphomimetic (from $P_7$–$P_{12}$) cardiomyocytes (Fig. 4K). These data suggest that the $Ca^{2+}$ removal efficiency of the phosphoresistant SERCA2 was being so sizeable that it further reduced the NCX role in the competition for cytosolic $Ca^{2+}$ removal, corroborating results from Fig. 4D.

Adding that basal cytosolic $Ca^{2+}$ concentration was significantly reduced in phosphoresistant cardiomyocytes (Fig. 4E), these data extended previous HEK and hiPSC-CM data to cardiomyocytes showing that SERCA2 activity is controlled by its phosphorylation state on serine 663 residue in adult cardiac cells and that the phosphoresistant rS663A mutant augments SR/ER $Ca^{2+}$ content and reduces cytosolic $Ca^{2+}$ content.

Due to the importance of mitochondria in cardiomyocyte function, we next evaluated mitochondrial functions in intact SERCA2-rescued cardiomyocytes. No difference was observed between the three mutants at baseline neither in reactive oxygen species production (Supplementary Fig. 3H), nor in the mitochondrial membrane potential (Supplementary Fig. 3I), indicating that isolated cardiomyocytes were healthy with metabolically active mitochondria. In agreement, the three mutants showed a comparative basal respiratory competence of cardiac mitochondria (Supplementary Fig. 3J). However, challenging mitochondria

towards their maximal respiratory capacity showed an altered respiration function in the rS663E mutants versus the rS663A ones, suggesting that under supra-physiological or pathological conditions, the impaired cytosolic $Ca^{2+}$ homeostasis predisposes mitochondria/cells to dysfunction dependent of the phosphorylation of SERCA2 at S663.

## Phosphorylation state of SERCA2 at S663 regulates in vivo myocardial infarct size

Our above results led us to hypothesize that limitation of SERCA2 phosphorylation on S663 residue at reperfusion could provide protection against reperfusion injury, via the activation of the $Ca^{2+}$ reuptake into SR/ER which confers detoxification of the cytosolic and mitochondrial $Ca^{2+}$ overload during reperfusion. For this purpose, we used our transgenic mouse models combining the cardiac-inducible SERCA2-KD mice with the AAVs gene transfer, to express either a wild type (rWT), a phosphoresistant (rS663A) or a phosphomimetic (rS663E) rescued form of SERCA2 into the heart (Fig. 5A). All analyzed mice presented a cardiomyocyte-specific excision of the endogenous *Serca2* and a SERCA2 mutant rescue (Supplementary Fig. 4A, B). Transgenic mice underwent 60 minutes ligation of the left anterior descending artery followed by 24 h reperfusion (Fig. 5A). The areas at risk (AR) were comparable among the three groups (Fig. 5B). When necrosis was plotted versus AR (Fig. 5C), most data points for rS663A group were below the rWT rescue regression line, indicating that for any size of AR, these hearts developed significantly smaller infarcts, averaging 21.5% vs. 32.8% of AR in rWT (Fig. 5D). Conversely, most data points for rS663E rescued mice were above the rWT regression line (Fig. 5C,E), indicating that the rS663E hearts aggravated the I/R injury as exhibited by a significantly higher infarct size, averaging 44.7% of AR (Fig. 5D). Additionally, it is worth noticing that the phosphomimetic rS663E mutant induced a higher mortality after I/R (27%) versus rWT (11%) and phosphoresistant rS663A mutant (8%).

To determine the underlying mechanisms of the protection afforded by the S663A rescue, we analyzed the effect on known cardioprotective signaling pathways: no change was observed in the phosphorylation level of ERK and STAT3 after I/R between the three groups (Fig. 5F, G). Interestingly, following an in vivo ischemia-reperfusion, we measured a significantly decreased phosphorylation of PLN at both S16 and T17 in the r663E cardiomyocytes compared to either rWT or rS663A cells in the AR (Fig. 5H), with no significant effect on the interaction between SERCA and PLN (Fig. 5I). We next evaluated the functions of the mouse cardiomyocytes isolated from the AR after the in vivo ischemia-reperfusion insult in a new set of WT-rescued or SERCA2-phosphoresistant mice to specifically address the cardioprotective effect

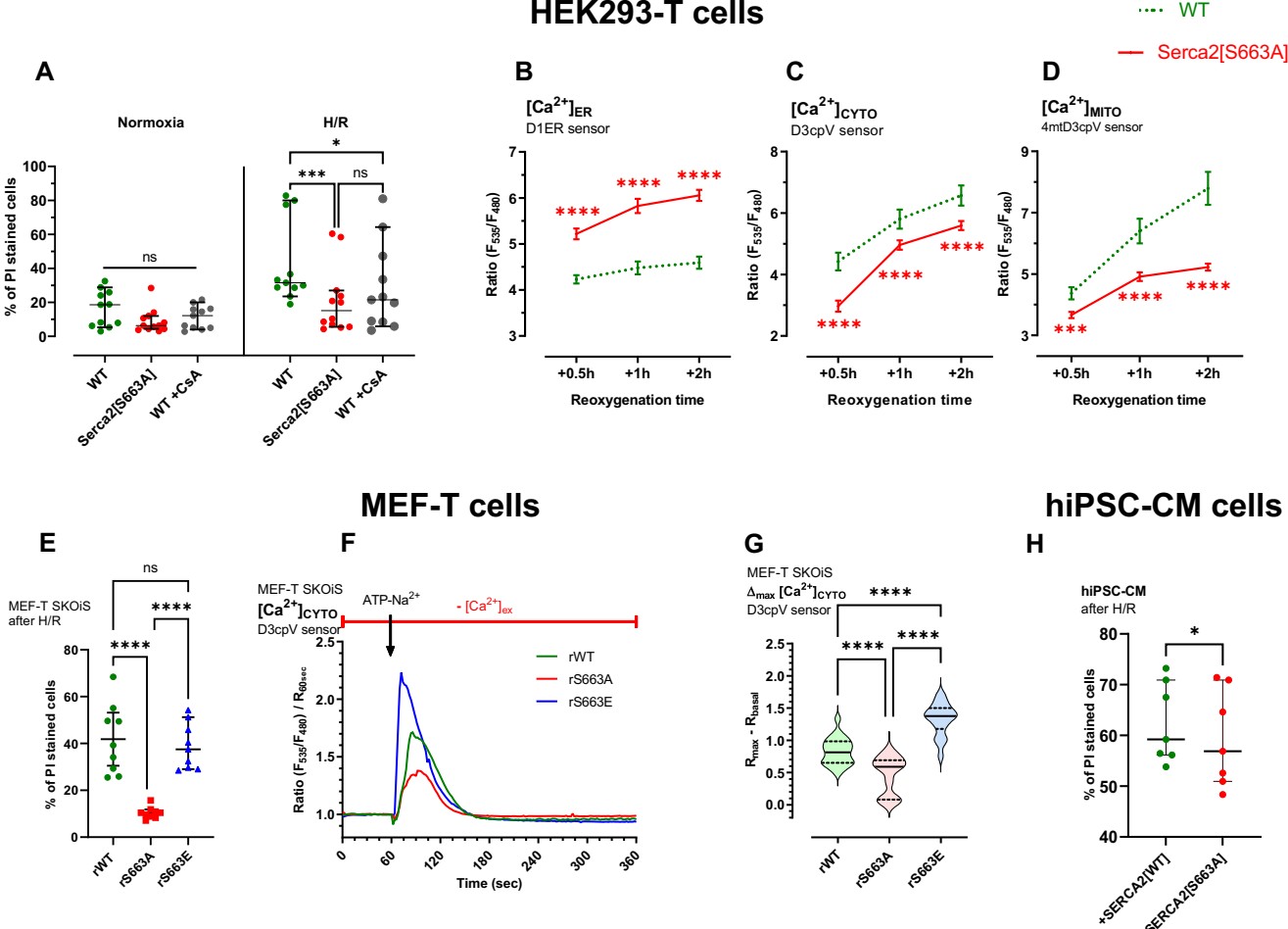

**Fig. 3 | Phosphoresistance of SERCA2 at S663 confers in vitro protection against H/R injury. A** Cell death evaluated by flow cytometry with propidium iodide (PI) in normoxic and hypoxic WT ($N = 11$) and Serca2[S663A] ($N = 12$) groups. Cyclosporin A (CsA) ($N = 11$) was used as a positive control. Median with a 95% confidence interval from 4 independent days is shown and two-way ANOVA followed by Tukey's multiple comparisons test versus WT condition was used to assess significance (ns $p \geq 0.05$, *$p = 0.0119$, ***$p = 0.004$. **B** Kinetic curves of [Ca$^{2+}$]$_{ER}$ (endoplasmic reticulum) ratio measured with D1ER probe at 0.5 h/1 h/2 h reoxygenation of HEK293-T wild-type (WT) (dotted green curve, $N = 103/86/89$ for respective reperfusion time, on 4 experimental days) and Serca2[S663A] phosphoresistant mutant (continuous red curve, $N = 90/95/95$ for respective reperfusion time, on 4 experimental days) cells exposed to 18 h hypoxia (1%). Mean with a 95% confidence interval is shown and two-tailed unpaired $t$ test with Welch's correction was used to assess significance (****$p < 0.0001$). **C** Kinetic curves of [Ca$^{2+}$]$_{CYTO}$ (cytosol) ratio measured with D3cpV probe at 0.5 h/1 h/2 h reoxygenation of HEK293-T wild-type (WT) (dotted green curve, $N = 71/44/36$ for respective reperfusion time, on 4 experimental days) and Serca2[S663A] phosphoresistant mutant (continuous red curve, $N = 64/73/72$ for respective reperfusion time, on 4 experimental days) cells exposed to 18 h hypoxia (1%). Mean with a 95% confidence interval is shown and two-tailed unpaired $t$ test with Welch's correction was used to assess significance (****$p < 0.0001$). **D** Kinetic curves of [Ca$^{2+}$]$_{MITO}$ (mitochondria) ratio measured with 4mtD3cpV probe at 0.5 h/1 h/2 h reoxygenation of HEK293-T wild-type (WT) (dotted green curve, $N = 73/55/47$ for respective reperfusion time, on 4 experimental days) and Serca2[S663A] phosphoresistant mutant (continuous red curve, $N = 72/70/97$ for respective reperfusion time, on 4 experimental days) cells exposed to 18 h hypoxia (1%). Mean with a 95% confidence interval is shown and two-tailed unpaired $t$ test with Welch's correction was used to assess significance (***$p = 0.009$, ****$p < 0.0001$). **E** Cell death evaluated by flow cytometry with propidium iodide (PI) in hypoxic MEF-T *Serca2* KO cell line, rescued with *Serca2* wild-type (rWT), [S663A] phosphoresistant mutant (rS663A) and [S663E] phosphomimetic mutant (rS663E). Mean with a 95% confidence interval from $n = 9$ experiments on 3 independent days is shown and two-way ANOVA followed by Tukey's multiple comparisons test versus rWT condition was used to assess significance (ns $p = 0.8029$, ****$p < 0.0001$). **F** Representative curves of normalized [Ca$^{2+}$]$_{CYTO}$ (cytosol) ratio signals over time measured with D3cpV probe in *Serca2 null* MEF-T cells and rescued with *Serca2* wild-type (rWT; green curve), [S663A] phosphoresistant mutant (rS663A; red curve) or [S663E] phosphomimetic mutant (rS663E; blue curve) cells in Ca$^{2+}$ free and 1 mM EGTA-containing buffer for one minutes prior to add 100 μM ATP-Na$^{2+}$. **G** Violin plots of normalized [Ca$^{2+}$] maximal increase after ATP-Na$^{2+}$ stimulation of rWT ($N = 22$ cells on 3 independent days), rS663A ($N = 23$ cells on 3 independent days) and rS663E ($N = 18$ cells on 3 independent days). Median with a 95% confidence interval is shown and one-way ANOVA followed by Tukey's multiple comparisons test was used to assess significance (****$p < 0.0001$). **H** Cell death evaluated by flow cytometry with propidium iodide (PI) after H/R in hiPSC-CM cells infected with SERCA2 wild-type [WT] ($N = 7$) or phosphoresistant [S663A] mutant ($N = 7$). Median with interquartile range, from $n = 7$ independent experiments, is shown and two-tailed paired Wilcoxon test was used to assess significance (*$p = 0.0313$).

afforded by the inhibition of SERCA2 phosphorylation. In front of the higher mortality induced by the phosphomimetic mutant rescue after I/R, this group was not included in the cardiomyocyte experiments for ethical concerns. A reduced mortality with preserved mitochondrial membrane potential and diminished resting cytosolic Ca$^{2+}$ level was measured in the S663A-rescued cardiomyocytes compared to the WT

rescue (Fig. 5J–L). Analysis of sarcomere shortening parameters further revealed an enhanced sarcomere contraction in the rS663A cells versus the rWT ones after the in vivo I/R (Fig. 5M–P). Altogether, these results support that the protection afforded by the S663A relies on the increased activity of SERCA2 and the ensuing prevention of cytosolic Ca$^{2+}$ overload, resulting in improved contraction.

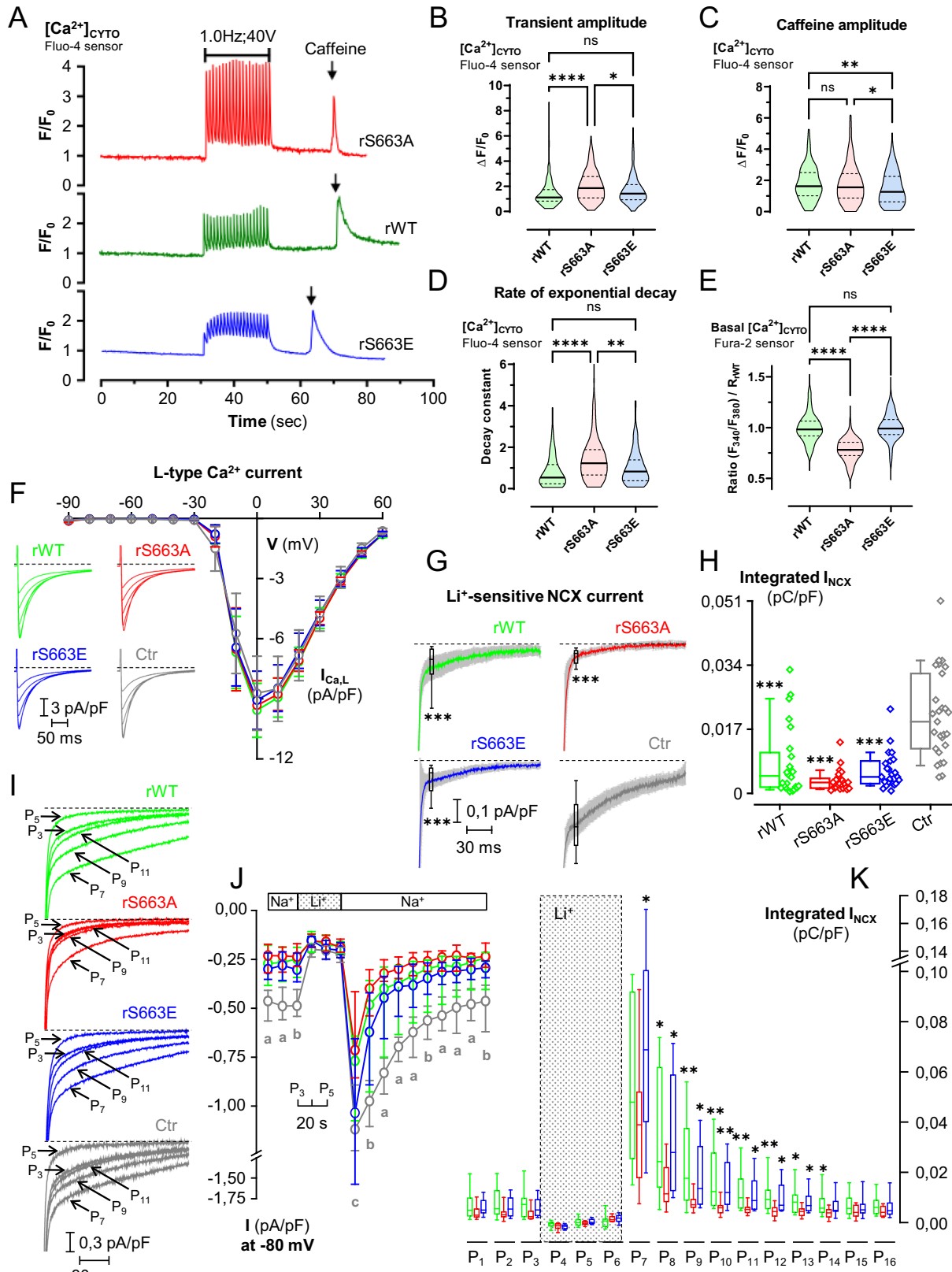

## Discussion

Here we show that the phosphorylation level of SERCA2 at serine 663 is a key regulator of $Ca^{2+}$ homeostasis and cell fate in heart. We demonstrated that the SERCA2 phosphorylation at serine 663 negatively regulates SERCA2 activity, inducing a decrease in SR/ER $Ca^{2+}$ content and a consequent $Ca^{2+}$ overload in both cytosol and mitochondria, which sensitizes cells to death (Fig. 6). We further show that mice expressing a phosphomimetic form of SERCA2 at S663 exacerbates the extent of myocardial infarction, reinforcing the relationship of this event with ischemic heart disease. It thus seems plausible that heart failure patients may suffer from a SERCA2 pump defect linked to its phosphorylation state since we identified a significant increase of the

**Fig. 4 | S663A phosphorylation modulates Ca²⁺ signaling and function in mouse cardiomyocytes. A** Typical curves of $[Ca^{2+}]_{CYTO}$ (cytosol) $F/F_0$ ratio signals over time measured with Fluo-4, in rWT (green curve), rS663A (red curve) and rS663E (blue curve) rescued cardiomyocytes, in 2 mM Ca²⁺ electro-stimulation buffer for 30 s, followed by 20 s electro-stimulation at 1.0 Hz and 40 V, and prior to 5 mM caffeine stimulation at time 60 s. **B** Violin plots of the transient amplitude during electro-stimulation in rWT ($N = 184$ cells on 5 experimental days), rS663A ($N = 253$ cells on 5 experimental days) and rS663E ($N = 260$ cells on 6 experimental days). Median with interquartile range is shown and one-way ANOVA followed by Tukey's multiple comparisons test was used to assess significance (ns $p = 0.0666$, **$p = 0.0014$, ****$p < 0.0001$). **C** Violin plots of the $[Ca^{2+}]$ maximal increase in rWT ($N = 214$ on 5 experimental days), rS663A ($N = 164$ cells on 5 experimental days) and rS663E ($N = 257$ cells on 6 experimental days). Median with interquartile range is shown and one-way ANOVA followed by Tukey's multiple comparisons test was used to assess significance (ns $p = 0.9759$, *$p = 0.0241$, **$p = 0.0063$). **D** Violin plots of the rate of exponential decay after caffeine stimulation in rWT ($N = 94$ cells on 5 experimental days), rS663A ($N = 118$ cells on 5 experimental days) and rS663E ($N = 145$ cells on 6 experimental days). Median with interquartile range is shown and one-way ANOVA followed by Tukey's multiple comparisons test was used to assess significance (ns $p = 0.0666$, **$p = 0.0014$, ****$p < 0.0001$). **E** Violin plot of basal $[Ca^{2+}]_{CYTO}$ (cytosol) ratio signals measured with the ratiometric chemical Ca²⁺ indicator Fura-2, in rWT ($N = 111$ cells on 3 independent days), S663A ($N = 66$ cells on 3 independent days) and rS663E ($N = 69$ cells on 3 independent days). Median with interquartile range is shown and one-way ANOVA followed by Tukey's multiple comparisons test was used to assess significance (ns $p = 9012$, ****$p < 0.0001$). **F** Current–voltage relationships of peak $I_{Ca,L}$ normalized to membrane capacitance from rWT (green curve; $N = 16$), rS663A (red curve; $N = 15$), rS663E (blue curve; $N = 15$) and non-transgenic control (Ctr; gray curve; $N = 17$) cells isolated from 3 mice in each group. Data are expressed as mean ± CI 95%. Inserted on the left, representative traces of L-type Ca²⁺ current in rWT, rS663A, rS663E and Ctr cardiomyocytes during depolarizing steps spaced 10 mV apart and varying between 0 and +40 mV (uppermost traces) from a holding potential of −80 mV. **G** Average traces (±CI 95%) of $I_{NCX}$ measured as the Li⁺-sensitive slow tail inward current recorded every 10 s when polarizing the cell to −80 mV after it has been depolarized 20 ms to −50 mV (to inactivate the fast sodium current), and then 30 ms to +10 mV (to activate $I_{Ca,L}$), from rWT ($N = 24$), rS663A ($N = 24$), rS663E ($N = 23$) and Ctr (C57Bl6J mice $N = 27$) cells isolated from 6, 7, 5 and 6 mice, respectively. Box (with median and interquartile range) and whiskers (10-90%) showing $I_{NCX}$ densities measured after 20 ms of repolarization to −80 mV have been added on every trace. (***$p < 0.001$ vs Ctr). **H** Panel **G** corresponding boxes (with median and interquartile range) and whiskers (10−90%) with dot plots showing $I_{NCX}$ densities measured as the integrated form of inward currents (135 ms integration starting 15 ms after the onset of repolarization). Expressed as mean ± CI 95%, membrane capacitances in pF were: rWT: 141.1 ± 11.9, $N = 40$; rS663A: 157.9 ± 14.7, $N = 39$; rS663E: 158.5 ± 15.3, $N = 38$, Ctr: 156.5 ± 13.6, $N = 44$. ***$p < 0.001$ indicates on **G** and **H** statistically significant differences compared to Ctr using the Kruskal−Wallis test followed by Dunn's multiple comparisons test. **I** Representative records of inward tail current normalized to membrane capacitance recorded at −80 mV from rWT, rS663A, rS663E and Ctr cardiomyocytes in Na⁺ (P3) and Li⁺ (P5) external solution and after return in the Na⁺ external solution (P7, P9 and P11). **J** Time course of inward tail current densities measured 20 ms after the onset of the repolarization to −80 mV from rWT ($N = 19$), rS663A ($N = 18$), rS663E ($N = 20$) and Ctr ($N = 21$) cells isolated from same mice as in (**F**) and (**G**). First placed in external Na⁺ ($P_1$–$P_3$), the cells were then exposed to external Li⁺ ($P_4$–$P_6$), before being returned to external Na⁺ (from $P_7$ to $P_{16}$), Data are presented as median with interquartile range. The Kruskal−Wallis test followed by Dunn's multiple comparisons test was used, and only differences with Ctr were reported as significant. a, $p < 0.01$ at least for the 3 mutants versus Ctr, b, $p < 0.05$ at least for the 3 mutants versus Ctr, and c, $p < 0.05$ only for the rS663A mutant versus Ctr. **K** Panel **J** corresponding integrated NCX density currents from rWT, rS663A and rS663E mutant cardiomyocytes displayed in Na⁺ ($P_1$–$P_3$) and Li⁺ ($P_4$–$P_6$) external solution and after return in the Na⁺ external solution (from $P_7$ to $P_{16}$). Data are presented as box (median with interquartile range) and whiskers (10−90%). The Kruskal−Wallis test followed by Dunn's multiple comparisons test was used and only differences with rS663A were reported as significant (P1: $p = 0.0702$, P2: $p = 0.1466$, P3: $p = 0.0573$, P4: $p = 0.0964$, P5: $p = 0.4200$, P6: $p = 0.1806$, P7: $p = 0.0401$, P8: $p = 0.0113$, P9: $p = 0.0054$, P10: $p = 0.0007$, P11: $p = 0.0039$, P12: $p = 0.0033$, P13: $p = 0.0175$, P14: $p = 0.0318$, P15: $p = 0.2819$ and P16: $p = 0.1083$, vs rS663A). Dash lines on **F**, **G** and **I** indicate zero current.

SERCA2 phosphorylation level at serine 663 in these patients. Our study extends previous findings for the defect of SERCA2 in cardiac diseases[11], nevertheless remaining the first to identify serine 663 phosphorylation level as a key event in Ca²⁺ dyshomeostasis and post-ischemic cardiac disease progression.

On the other hand, we demonstrated that preventing SERCA2 phosphorylation at serine 663 provides in vivo protection against reperfusion injury. Interestingly, the cardioprotection afforded by the inhibition of SERCA2 phosphorylation did not depend on the activation of the RISK and SAFE signaling pathways, but on the detoxification of the cytosolic Ca²⁺ overload, as shown by the reduced cytosolic Ca²⁺ level in cardiomyocytes after in vivo I/R. Therefore, the cardioprotective effect of the phosphoresistant SERCA2[S663A] mutant may be mainly due to the increase of Ca²⁺ uptake into SR/ER, which consequently prevents both cytosolic and mitochondrial Ca²⁺ overload at reperfusion, limiting cell death and subsequent cardiac damages after ischemia-reperfusion while improving cardiomyocyte contractility (Fig. 6). Our study focused on the role of SERCA2 phosphorylation on cell death; however, one could wonder whether the decreased cell death and improved cardiomyocyte contractility afforded by SERCA2 phosphorylation inhibition could lead to improved post-ischemic myocardial function, paving the way for a future study. In parallel, despite the debated involvement of NCX and LTCC to act synergistically to prime and to trigger Ca²⁺ for ECC[38–41], our results demonstrated that the acceleration of Ca²⁺ uptake by SERCA2 improves Ca²⁺ transients in phosphoresistant SERCA2[S663A] mutant, which may contribute to a better excitation-contraction coupling in cardiomyocytes in response to reperfusion injury, without modification of NCX or LTCC functions in our experimental conditions. The fact that overexpressing SERCA2 pumps leads to better SR Ca²⁺ transport without changing these sarcolemma electrophysiological properties has been already reported[42,43] and we admittedly recognize that it is a limitation of our rescued model.

Nevertheless, despite this bias and since all mutants are overexpressed, all our results converge to show that the phosphoresistant mutant has better cytosolic Ca²⁺ removal skills to supplant NCX in performing this task. Thus, the originality of our work is to propose the use of an improved SERCA2 pump to manage episodes of Ca²⁺ overload, without relying on an overexpression that could exacerbate arrhythmias[44].

Because SERCA2-mediated Ca²⁺ transport is responsible for the efficient cardiac muscle relaxation and maintaining SR/ER Ca²⁺ stores, the myocardial expression and activity of SERCA2 pump have been studied extensively, both using experimental animal models and heart tissues from human end-stage HF. These studies collectively showed that SERCA2 mRNA and protein levels were decreased in heart diseases[45,46]. However, in some studies, the SERCA2 expression level was found to be unaltered[47,48] despite a decrease in SR/ER Ca²⁺ transport function. These results therefore support that other phenomena are involved to explain the modulation of SERCA2 activity, notably post-translational modifications of SERCA2. Our results indeed show that SERCA2 presents no less than 11 predicted sites for potential phosphorylation by the GSK3β kinase, and that the steady-state SERCA2-GSK3β proximity is strongly enhanced during hypoxia-reoxygenation. One may question whether the possible interaction between the two proteins is a stable complex or a transient kiss-and-run event. Such mechanism remains to be investigated in-depth in future studies.

In addition to the structural predisposition of SERCA2 to be phosphorylated at serine 663 (Fig. 1G), our mass spectrometry approach revealed also a significant increase in SERCA2 phosphorylation at serine 378 in ischemic mouse hearts. Since this serine residue was not sorted out by the predictive analysis of phosphorylation sites by GSK3β, we chose to focus our analysis only on the serine 663. Nevertheless, it will be interesting to investigate in-depth the role and function of this GSK3β-independent phosphorylation site of SERCA2 during cardiovascular

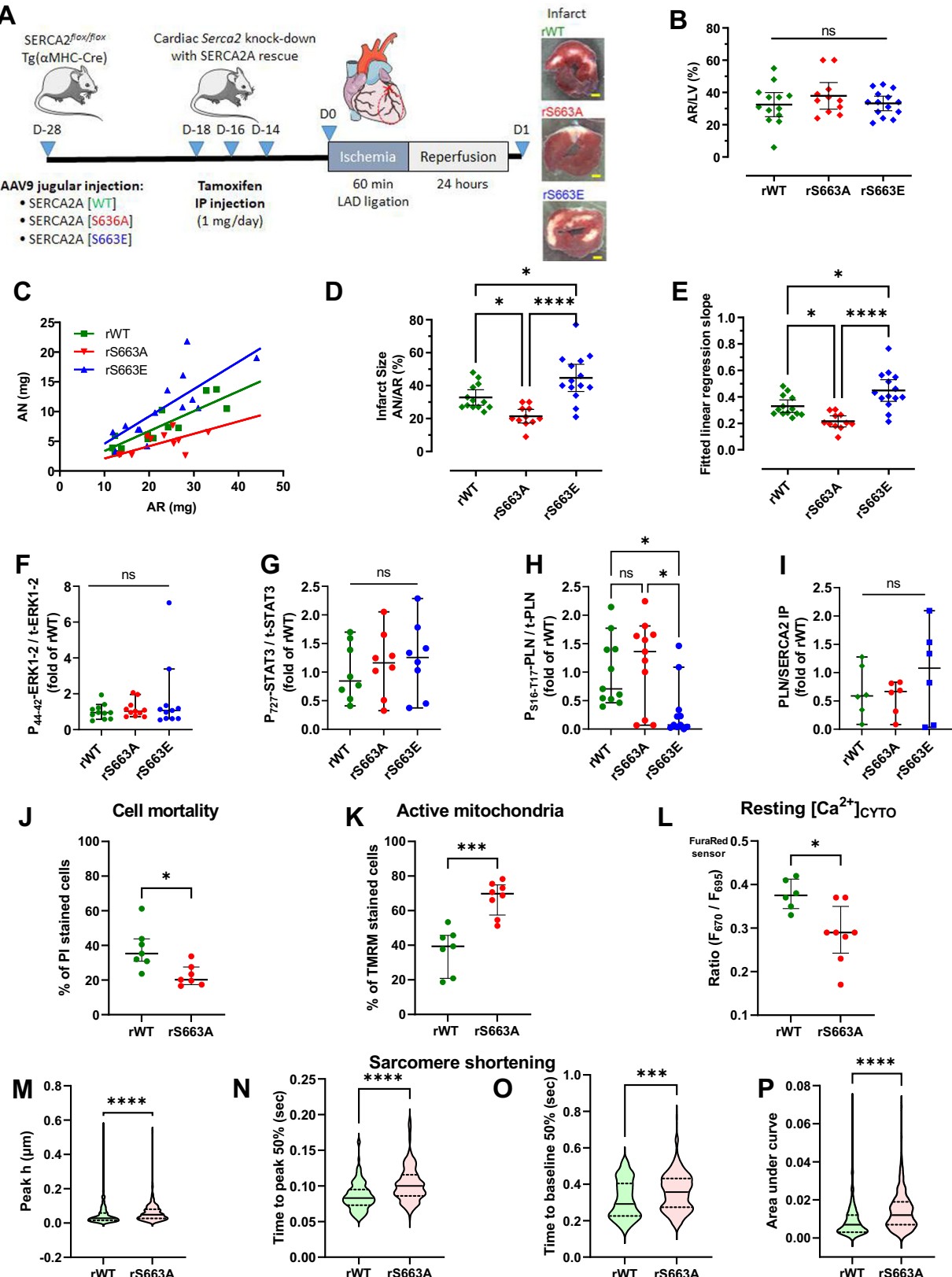

diseases, potentially opening the path for a multi-targeted immunotherapy, even though phosphorylation on serine 378 was not identified in the human samples, limiting its translational value. Parenthetically, other SERCA2 phosphorylation sites have been described in the past: these are serine 38[26] and threonine 484[25]. However, these residues were not identified by mass spectrometry phosphorylation

mapping of SERCA2 in our experimental conditions, although the peptides were indeed detected, suggesting that these residues are unlikely to be involved in the ischemic heart.

Interestingly, we observed a decreased PLN phosphorylation in the phosphomimetic SERCA2-rescued cardiomyocytes only after I/R, which could therefore further increase the SERCA2 activity and

**Fig. 5 | Phosphorylation state of SERCA2 at S663 regulates in vivo myocardial infarct size. A** Experimental design for in vivo gene therapy experiments with heart transverse section after TTC staining. Scale bar represents 500 μm. **B** Quantification of the area at risk (AR) expressed as percentage of left ventricle (LV) of SERCA2-KD mice rescued with SERCA2 -WT ($N = 13$), -S663A ($N = 11$) or -S663E ($N = 14$). Mean with a 95% confidence interval is shown and one-way ANOVA followed by Tukey's multiple comparisons test was used to assess significance (ns $p \geq 0.05$). **C** Scatterplot of AN over the AR of mice rescued with SERCA2 -WT ($N = 13$), -S663A ($N = 11$) or -S663E ($N = 14$). **D** Quantification of infarct size (AN) expressed as percentage of AR of mice rescued with SERCA2 -WT ($N = 13$), -S663A ($N = 11$) or -S663E ($N = 14$). Mean with a 95% confidence interval is shown and one-way ANOVA followed by Tukey's multiple comparisons test was used to assess significance (*$p = 0.0292$ rWT vs rS663A, *$p = 0.0144$ rWT vs rS663E, ****$p < 0.0001$). **E** Dot plot of fitted linear regression slope of AN/AR of rWT ($N = 13$), rS663A ($N = 11$) or rS663E ($N = 14$) SERCA2 rescued mice. Mean with a 95% confidence interval is shown and one-way ANOVA followed by Tukey's multiple comparisons test was used to assess significance (*$p = 0.0272$ rWT vs rS663A, *$p = 0.0136$ rWT vs rS663E, ****$p < 0.0001$). **F–I** Evaluation of cardioprotective signaling pathways in lysates of cardiac area at risks. **F** Quantification of western blotting for Phospho-ERK over total ERK1-2 in rWT ($N = 11$), rS663A ($N = 11$) and rS663E ($N = 11$) SERCA2 rescued mice. Mean with a 95% confidence interval, from $n = 3$ independent experiments, is shown and one-way ANOVA followed by Tukey's multiple comparisons test was used to assess significance (ns $p > 0.05$). **G** Quantification of western blotting for phospho-STAT3 over total STAT3 in rWT ($N = 8$), rS663A ($N = 8$) and rS663E ($N = 8$) SERCA2 rescued mice. Mean with a 95% confidence interval, from $n = 3$ independent experiments, is shown and one-way ANOVA followed by Tukey's multiple comparisons test was used to assess significance (ns $p > 0.05$) **H** Quantification of western blotting for phospho S16-T17-PLN over total PLN, expressed as fold of rWT. Mean with a 95% confidence interval, from $n = 3$ independent experiments, is shown and one-way ANOVA followed by Tukey's multiple comparisons test was used to assess significance (ns $p = 0.8195$, *$p = 0.0102$ rS663A vs rS663E, *$p = 0.0424$ rWT vs rS663E). **I** Quantification of the interaction of PLN with SERCA2 determined by co-immunoprecipitation on lysates from cardiac area at risks in rWT ($N = 6$), rS663A ($N = 6$) and rS663E ($N = 6$) SERCA2 rescued mice. Mean with a 95% confidence interval, from $n = 3$ independent experiments, is shown and one-way ANOVA followed by Tukey's multiple comparisons test was used to assess significance (ns $p > 0.05$). **J–L** Evaluation of the functions of the mouse cardiomyocytes isolated from the AR after the in vivo ischemia-reperfusion insult. **J** Cell death evaluated by flow cytometry with propidium iodide (PI) in rWT ($N = 7$) and rS663A ($N = 7$) SERCA2 rescued mice. Median with interquartile interval is shown and two-tailed unpaired $t$ test with Welch's correction was used to assess significance (*$p = 0.0104$). **K** Mitochondrial membrane potential evaluated with TMRM in rWT ($N = 7$) and rS663A ($N = 8$) SERCA2 rescued mice. Median with interquartile interval is shown and two-tailed unpaired $t$ test with Welch's correction was used to assess significance (***$p = 0.0002$). **L** Resting cytosol Ca$^{2+}$ evaluated with the ratiometric FuraRed sensor in rWT ($N = 6$) and rS663A ($N = 8$) SERCA2 rescued mice. Median with interquartile interval is shown and two-tailed unpaired $t$ test with Welch's correction was used to assess significance (*$p = 0.0105$). **M–P** Quantification of sarcomere shortening in mouse cardiomyocytes isolated from the AR after the in vivo ischemia-reperfusion insult measured with the software IonWizard; Ionoptix system. **M** Evaluation of peak h in μm in rWT ($N = 135$) and rS663A ($N = 245$) SERCA2 rescued cardiomyocytes. Median with interquartile interval is shown and two-tailed unpaired $t$ test with Welch's correction was used to assess significance (****$p < 0.0001$). **N** Evaluation of time to peak 50% in rWT ($N = 135$) and rS663A ($N = 245$) SERCA2 rescued cardiomyocytes. Median with interquartile interval is shown and two-tailed unpaired t test with Welch's correction was used to assess significance (****$p < 0.0001$). **O** Evaluation of time to baseline 50% in rWT ($N = 133$) and rS663A ($N = 244$) SERCA2 rescued cardiomyocytes. Median with interquartile interval is shown and two-tailed unpaired $t$ test with Welch's correction was used to assess significance (***$p = 0.0003$). **P** Evaluation of area under curve in rWT ($N = 135$) and rS663A ($N = 242$) SERCA2 rescued cardiomyocytes. Median with interquartile interval is shown and two-tailed unpaired t test with Welch's correction was used to assess significance (****$p < 0.0001$).

aggravate the cytosolic Ca$^{2+}$ overload. Emerging studies are trying to decipher the molecular interplay of the modulation of SERCA2 by its possible subunits, PLN and sarcolipin[49,50]. SERCA2 phosphorylation on serine 663 may be part of this process and will deserve future research.

SERCA2 is the major isoform expressed in cardiomyocytes[35] and thus the focus of our study. Interestingly, the protein sequence alignment across human SERCA isoforms 1, 2, and 3 reveals that the serine 663 (S663) of SERCA2 corresponds to an alanine 664 (A664) in SERCA1 and to a glutamine 664 (E664) in SERCA3 (UniProt IDs: O14983, P16615 and Q93084 respectively). Moreover, as compared with SERCA2, while the SERCA1 isoform with A664 has been demonstrated to have a better ER Ca$^{2+}$ pumping rate[51], the SERCA3 isoform with E664 has been established to have a lower ER Ca$^{2+}$ pumping rate as compared with SERCA2[52]. Although these observations require further investigations, these data suggest that our S663-SERCA2 phosphoresistant and phosphomimetic mutants partly reflect the natural differences in Ca$^{2+}$ pumping observed between the human SERCA isoforms.

In summary, more than just proposing the activation of SERCA2 activity as potential therapies[53], our results specify that preventing SERCA2 phosphorylation at serine 663 reduces Ca$^{2+}$ dyshomeostasis, cell death and infarct size, key features of ischemia-reperfusion injury. In the aging population, the incidence and prevalence of heart failure are increasing consecutively to ischemic heart diseases. We believe that our study paves the way for prospective clinical trials to evaluate the potential cardiovascular strategies to prevent specific phosphorylation of SERCA2 at serine 663.

## Methods

### Patient sample collection

Our study complies with Tier 1 characteristics for Biospecimen reporting for improved study quality (BRISQ) guidelines [https://acsjournals.onlinelibrary.wiley.com/doi/10.1002/cncy.20147]. Failing heart tissues were processed at the Louis Pradel hospital, Hospices Civils de Lyon and obtained with informed consent from explanted heart of patient admitted for heart transplantation (with severe reduction of ejection fraction (DCM, ICM)). All donor sample were full-thickness myocardial biopsies from the left ventricle. Heart biopsies were excised, rinsed in cold saline, and samples were flash-frozen and stored at −80 °C before tissue dissociation for proteomic studies. Non-failing heart samples and associated data were obtained with informed consent from Tissu-Tumorotheque Cardiobiotec (CRB-HCL, Hospices Civils de Lyon Biobank, BB-0033-00046 after Research Ethic Committee approval AC-2019-3464 and AC-2020-3918. Collected biopsies from male and female patients, were stored at −80 °C before tissue dissociation for proteomic studies.

### Plasmid constructs

**Phosphoresistant and -mimetic mutants of hSERCA2A.** For substitution of serine 663 residue of hSERCA2A, the CMV-*hSERCA2A* plasmid (Addgene #75187)[54] was mutated by PCR using mutant primers and Q5 site-directed mutagenesis kit (New England Biolabs, E0554) according to the manufacturer's instructions. Mutant primers were designed to modify codon coding for serine 663 (i.e. TCC) into phosphoresistant alanine (i.e. GCG), or into phosphomimetic glutamic acid (i.e. GAA).

The primers used for creating nucleotide substitutions were S663A/for: 5'.ACTCAACCCCgcgGCCCAGCGAG.3', S663E/for: 5'ACTCAACCCCgaaGCCCAGCGAG.3' and a common hSERCA2A(1976)/rev: 5'.TCATCAAACTCCCGGCCT.3' (mutated nucleotides are in lowercase). Created plasmids DNA sequences (CMV-*hSERCA2A[S663A]* and CMV-*hSERCA2A[S663E]*) were confirmed to possess the targeted mutations by Sanger sequencing.

**All-in-one CRISPR/Cas9-nuclease and gRNA vector.** For substitution of serine 663 residue (TCC codon) of SERCA2 protein into phosphoresistant alanine (GCG codon), a gRNA was designed to anneal sense strand downstream targeted locus in the *ATP2A2* (*SERCA2*;

**Fig. 6 | Mechanistic role of SERCA2 phosphorylation at S663 in ischemic and healthy/protected hearts.** In healthy heart, during excitation-contraction coupling (ECC) induced after depolarization of the sarcolemma, a small amount of $Ca^{2+}$ enters to the sarcoplasm through the voltage sensitive L-type calcium dihydropyridine channels (LTCC) of the T-tubule, targeting a rapid and large amount of $Ca^{2+}$ release (systolic $Ca^{2+}$) from inside the sarcoplasmic reticulum (SR/ER) through the $Ca^{2+}$ ryanodine channels (RyR), which subsequently activates myofilaments (MF) for muscle contraction. Relaxation occurs when SR $Ca^{2+}$ ATPase (SERCA2) reuptakes $Ca^{2+}$, which is regulated by the balance of the phosphorylation level of SERCA2 at serine 663 (S663) via the GSK3β, lowering cytosolic $Ca^{2+}$ concentration in combination with $Ca^{2+}$ extrusion via $Na^+$–$Ca^{2+}$ exchanger (NCX) working in reverse mode. Mitochondria (Mito) also participate taking and extruding $Ca^{2+}$ from the cytosol during $Ca^{2+}$ cycle. During ischemic heart disease, the level of SERCA2 phosphorylation at S663 is increased, reducing the reuptake of $Ca^{2+}$ into SR/ER during the relaxation phase, which contributes to the increase of both intracellular ($Ca^{2+}_i$) and mitochondrial $Ca^{2+}$ ($Ca^{2+}_m$) overload, driving cells towards death. Conversely, preventing SERCA2 phosphorylation at reperfusion, enhances the $Ca^{2+}$ reuptake into SR/ER, increasing the SR/ER $Ca^{2+}$ content, which subsequently (1) reduces the intracellular and mitochondrial $Ca^{2+}$ content, and (2) improves excitation-contraction coupling of cardiomyocytes, contributing to the protection and recovery of ischemic heart.

ENSG00000174437) gene (base-pairing sequence: 5′.GTTCAGG-CAGGCGTCTCGCTggg---.3′) and cloned into pX330S-2 (Addgene #58778), expressing both Cas9-nuclease and gRNA, according to the protocol defined by Sakuma et al.[55]. Created plasmid DNA sequence (pX330S-2+gRNA-S663) was confirmed to possess the gRNA by Sanger sequencing.

### In silico analysis
The putative GSK3β binding sites on SERCA2 were predicted using NetPhos 3.1[33] [https://services.healthtech.dtu.dk/services/NetPhos-3.1/] and PhosphoSitePlus[34] [https://www.phosphosite.org/homeAction.action] public databases.

### Cell culture
HEK293-T (ATCC, CRL-3216), MEF-T cell lines and their derivatives were routinely cultured in Dulbecco's Modified Eagle Medium (Gibco) supplemented with 10% fetal calf serum (PAN-biotech), 100 μM non-essential amino acids (Gibco), 100 U/mL penicillin (Gibco), 100 μg/mL streptomycin (Gibco), 2 mM L-glutamine (Gibco), 1 mM sodium pyruvate (Gibco) and incubated at 37 °C in 5% $CO_2$ in a damp atmosphere.

Cells were regularly passaged by single-cell dissociation with 0.05% trypsin-EDTA (Gibco).

**Generation of transient transfectants.** All DNAs were transfected into cells using DharmaFECT Duo (Dharmacon, T-2010-03) according to manufacturer's instructions.

### Generation of *Serca2* transgenic cell line
To perform CRISPR/Cas9-genome editing, 200,000 HEK293-T cells were transfected with 500 ng of all-in-one Cas9-nuclease and gRNA plasmid (pX330S-2+gRNA-S663), 100 pmol of ssODN homology-directed repair (HDR) donor complementary to the anti-sense strand (5′.CGCCGC ATCGGCATCTTCGGGCAGGATGAGGACGTGACGTCAAAAGCTTTCACA GGCCGGGAGTTTGATGAACTCAACCCCgCgGCCCAGCGAGACGCCTG CCTGAACGCCCGCTGTTTTGCTCGAGTTGAACCCTCCC.3′; mutated nucleotides are in lowercase), and 50 ng of pcDNA4 (Invitrogen, V1020-20) Zeocin selectable plasmid. From 24 h after transfection, 1 μM SCR7 pyrazine (Sigma, SML1546) and 200 μg/mL Zeocin (Gibco, R25001) were added to promote HDR at Cas9-induced DSB[56], and to select transfected cells respectively. Single-cell-derived colonies were screened for

expected base substitution thanks to polymerase chain reaction (PCR) amplification of a 281 base pairs (bp) genomic sequence spanning targeted locus of *ATP2A2* (forward primer 5′.ACCAATCTGACCTT CGTTGG.3′ and reverse primer 5′.GAGGGTTCAACTCGAGCAAA.3′) followed by SacII digestion (NEB, R0157) and Sanger sequencing. ssODN-mediated HDR was design to insert a SacII restriction site that give two fragments of 227 and 54 bp after digestion that does not exist in wild-type genome. A validated clone was kept to generate HEK293-T SER-CA2[S663A] cell line used for this study.

### Isolation of Mouse Embryo Fibroblasts (MEFs) and generation of stable transfectants

Primary cultures of Mouse Embryo Fibroblasts (MEFs) were isolated from 11.5- to 13.5 days post coïtum mouse embryos from genetically engineered mouse strains *Serca2*[flox/flox] [36]. In brief, carcasses without head and viscera were dissect, then mechanical and enzymatic dissociation with 0.25% trypsin-EDTA (Gibco) was done to obtain a single-cell suspension which was plated onto culture dish in culture medium. MEFs were immortalized thanks to a large T-antigen plasmid (Addgene #18922), then they were selected with 1 μg/mL Puromycin (Sigma, P8833) and cloned to obtain a MEF-T cell line. For generation of cells stably expressing a rescued *SERCA2A* cDNA in an *Serca2*-null background, called SKOiS cells (for *Serca2 null* and induction of *SERCA2A*), MEF-T cell line was co-transfected with 0.2 μg of pPGK-nlsCRE-hygro, kindly provided by Dr. Pierre Savatier's lab, and 2.0 μg of PvuI (NEB, R0150) linearized CMV-*hSERCA2A* rescue plasmid (wild-type (Addgene #75187)[54], [S663A] mutant or [S663E] mutant), and stable transfectants were selected with 500 μg/mL G418 (Sigma, A1720) and 40 μg/mL Hygromycin B (Roche, 10843555001).

### hiPS cell culture and cardiomyocytes differentiation

hiPS control cell line (named *AG08C5*) was generated from primary fibroblasts (Coriell, ref. AG08498) using lentiviral infection with a polycistronic vector expressing OCT4, KLF4, SOX2 and c-MYC (OKSM vector, Millipore). *AG08C5* hiPSC line is maintained by the iPS-PGNM plateform ((https://pgnm.inmg.fr/plateformes-inmg/) [https://pgnm. inmg.fr/plateformes-inmg/] and was declared at the French minister of health (*CODECOH DC-2022-5055*). *AG08C5* hiPSC-derived cardiomyocytes (*AG08C5-CM*) were generated as previously described[57] with some modifications. Briefly, hiPSCs are plated on tissue culture dishes (Sarsted, 83.3901.500) coated with Matrigel® (Corning®, CLS356231) and amplified in mTeSR Plus medium (STEMCELL technologies, 100-0276). hiPSCs were then treated with 10 μM ChIR 99021 (C6556, LC laboratories®) in RPMI 1640 medium (Thermo Fisher Scientific) supplemented with B27 minus insulin (Thermo Fisher Scientific, A1895601) for three days, renewed after 2 days. hiPSCs are then treated with 2 μM C59-Wnt antagonist (Abcam, Ab142216) in RPMI 1640 medium supplemented with B27 minus insulin. After five days, cells are switch in RPMI 1640 medium supplemented with B27 minus insulin, renewed every alternate days till day 9. Medium is then replaced by RPMI 1640 plus B27 supplement (17504-044, Thermo Fisher Scientific) and changed on every alternate day for 20 days. *AG08C5-CM* are then removed from tissue culture dishes using TrypLE express (Thermo Fisher scientific, 12604013) and plated on glass coverslips coated with 40 μg/mL laminin (Sigma Aldrich, L2020) for at least 5 days before calcium flux experiments. hiPSC-CMs were infected after 5 days of culture once confluent monolayers had formed. Cells were exposed to viral dose MOI of 100,000 of AAV9[58] for a total of 2 days at 37 °C and fresh medium was changed every 48 h up to the sixth day for analysis[59].

### Simulated hypoxia/reoxygenation (H/R)

For HEK293-T and MEF-T cell lines, simulated H/R was performed using adapted protocol from Zervou et al.[60]. In brief, cells were plated out onto 12 well plates (550,000 for HEK293-T and 280,000 for MEF-T). One day after plating cells were exposed to 1% $O_2$, in ischemic-mimetic buffer (composition in mM: 125.0 NaCl, 8.0 KCl, 1.2 $KH_2PO_4$, 1.25 $MgSO_4$, 1.2 $CaCl_2$, 6.25 $NaHCO_3$, 5.0 sodium L-lactate, 20.0 HEPES; pH 6.6) for 18 h (HEK293-T) or 3h30 (MEF-T), followed by 4 h of reoxygenation in normoxic buffer (composition in mM: 110.0 NaCl, 4.7 KCl, 1.2 $KH_2PO_4$, 1.25 $MgSO_4$, 1.2 $CaCl_2$, 25.0 $NaHCO_3$, 15.0 glucose, 20.0 HEPES; pH 7.4). Drugs were added to normoxic buffer at the onset of reoxygenation [1 μM Cyclosporin A (CsA, Sigma, C3662)]. Normoxic group consisted of cells without the hypoxic stimulus. Cells were collected thanks to trypsin dissociation, re-suspended in 500 μL cold phosphate buffered saline (Gibco) and 1 μg/ml of propidium iodide (PI, Sigma, P4864) was added extemporaneously for cell death quantification by PI incorporation. For hIPSC-CM cells, simulated hypoxia/reoxygenation was performed using adapted protocol from Sebastiao et al.[61]. Briefly, infected hiPSC-CM cells (100,000 cells onto 12 well plates with a viral dose MOI of 200,000) underwent hypoxia by replacing RPMI + B27 medium by 1 ml of ischemic-mimetic solution (in mM: NaCl, 135; KCl, 8; $MgCl_2$, 0.5; $NaH_2PO_4$, 0.33; HEPES, 5.0; $CaCl_2$, 1.8; Na⁺-lactate, 20; pH 6.8) and by placing cells in a $N_2$ gaseous environment at 37 °C. After 5 h of hypoxia, reoxygenation was mimicked by addition of 1 ml of RPMI + B27 medium in normoxic condition for 5 more hours. The impact of H/R was evaluated regarding the percentage of dead PI-positive cells measured using the 561 nm laser on a LSRFortessa X-20 (BD-Biosciences) and then analyzed using FACSDiva 8.0.1 software (BD-Biosciences). Acquisition thresholds were set to select for cell population without debris and a live/dead gate set to 5% using a no-PI treated sample. A total of 10,000 cells per tubes were acquired in triplicate and then averaged.

### Ca²⁺ imaging

**Solutions for live cell imaging.** Cells were placed into buffers containing 140 mM NaCl, 5 mM KCl, 1 mM $MgCl_2$, 10 mM HEPES and 10 mM glucose, adjusted to pH 7.4; supplemented with 1 mM EGTA for $Ca^{2+}$-free buffer or with 2 mM $CaCl_2$ for $Ca^{2+}$-containing buffer. For electrostimulation, cells were placed into buffer containing 150 mM NaCl, 5.4 mM KCl, 2 mM $MgCl_2$, 10 mM HEPES, 1 mM glucose, 2.5 mM pyruvate, 5 mM creatine, 5 mM taurine and 2 mM $CaCl_2$. Drugs were added to buffers for stimulations [ATP-Na² (Sigma, A7699); ATP-$Mg^{2+}$ (Sigma, A9187); 1 μM ionomycin (Abcam, ab-120370); 100 μM cyclopiazonic acid (CPA, Sigma, C1530); 1 μM thapsigargin (TG, Sigma, T9033); 1 μM SB216763 (Sigma, S3442); 10 μM TDZD8 (Sigma, T8325); 5 mM caffeine (Sigma, C0750)].

**Ca²⁺ probes.** Genetically encoded $Ca^{2+}$ probes D3cpV (Addgene #36323)[62], 4mtD3cpV (Addgene #36324)[62] and D1ER (Addgene #36325)[63] were transiently transfected into cells 48 h before monitoring $Ca^{2+}$ levels and dynamics in different cellular compartments, thanks to their FRET-based ratiometric properties. Chemical intracellular $Ca^{2+}$ indicators, ratiometric Fura-2 AM (Invitrogen, F1221) or non-ratiometric Fluo-4 AM (Invitrogen, F14201), were loaded into the cells by incubating 5 μM for 30 minutes at 20 °C or 37 °C respectively.

**Image acquisition and analysis.** Fura-2 AM, D3cpV, 4mtD3cpV and D1ER measurements were performed at 37 °C using a wide-field Leica DMI6000B microscope equipped with a 40× oil-immersion objective and an ORCA-Flash4.0 digital camera (HAMAMATSU). The fluorescent ratios were analyzed with MetaFluor 6.3 (Universal Imaging) after removing background fluorescence.

### Immunoblotting

**Immunoblotting (IB).** For IB, frozen cell pellets were lysed in RIPA buffer complemented with protease and phosphatase inhibitors. Protein lysates were then cleared by centrifugation (17,000 × *g* for 20 min). After SDS-PAGE and electroblotting on polyvinylidene fluoride, the membranes were incubated with specific primary antibodies rabbit anti-SERCA2 (Cell Signaling, 4388; 1/500); mouse anti-GSK3β

(Abcam, ab-93926; 1/4000); mouse anti-GSK3β-phospho S9 (Abcam, ab-54537; 1/1000); rabbit anti-GSK3β-phospho Y216 (Abcam, ab-75745; 1/1000); mouse anti-PLN (Abcam, ab-2865; 1/4000); rabbit anti pS16-T17-PLN (Cell signaling 8496; 1/4000); rabbit anti-GAPDH (Santa Cruz, sc-25778; 1/2000); anti-βactin (Sigma, A3854; 1/10,000); rabbit anti NCX1 (Abcam ab177952; 1/1000); rabbit anti pS727 STAT3 (Adcam ab86430); antimouse STAT3 (Santacruz, sc8019; 1/1000); antirabbit p44/42ERK (Cell signaling, cs9102; 1/1000); antimouse total ERK1-2 (MAB1576; 1/1000). Blots were incubated with horseradish peroxidase (HRP)-coupled sheep antimouse IgG (GE Healthcare, NA931VS; 1/10,000) and (HRP)-coupled goat antirabbit IgG (GE Healthcare, NA934VS; 1/10,000), and developed with Clarity Western ECL Substrate (BioRad, 1705060).

**Immunoprecipitation (IP).** For co-IP of SERCA2 related proteins, 200 μg of protein lysate was combined with 5 μL of anti-SERCA2 antibody (Cell Signaling, 4388) in a total volume of 200 μL with binding buffer. Tris-buffered saline (TBS) containing 1 mM EDTA and 1.0% Triton X-100 detergent used as binding/wash buffer. The antibody-antigen reaction was incubated overnight at 4 °C with end-over-end mixing. For each reaction, 50 μL of suspended *PureProteome Protein G Magnetic Bead* slurry (Millipore, LSKMAGG10) was washed twice with 500 μL of wash buffer by vigorously vortexing for 10 s, collecting the beads on the magnet and removing the buffer with a pipette. The beads were resuspended with 100 μL of wash buffer and added to the previously incubated antibody-antigen reaction. The mixture was incubated at 4 °C for 2 hours with continuous end-over-end mixing. The unbound fraction was discarded and, as described above, the beads were washed three times with 500 μL of wash buffer using the magnet to capture the beads. Denaturing elution was performed by adding 30 μL of reducing *2x Laemmli Sample Suffer* (BioRad, 1610737), containing 5% 2-mercaptoethanol, to the beads and incubating at 95 °C for 10 min. Elution was collected using the magnetic stand and saved for IB analysis, as described above.

## FRET analysis

**3sFRET setup.** Fluorescence resonance energy transfer (FRET) experiments were performed with the three channel sensitized emission method on living cells plated on a glass coverslip. Image acquisition was performed on a Nikon A1Rplus confocal microscope. A 405 nm laser line was preferred to a 458 nm one to limit acceptor bleed-through. The excitation light path was composed of: a 405/514/561/638 dichroic mirror, a 450/50 band pass in front of the first photomultiplier, a 50/50 beam splitter prior to a 525/50 band pass in front of the first GaASP and a 560 Long pass prior to a 525/50 band pass in front of the second GaASP. Sensitivity of the two Gasps and their gain/offset settings were adjusted with the 514 nm argon laser when exciting acceptor-alone transfected cells. Sensitivity of FRET and donor light paths were adjusted (same transmission power, different gain) in order to obtain a ratio FRET/Donor = 0.5. Finally, the 514 nm laser power was adjusted. All settings were kept for all acquisitions. A ×60/1.4 NA oil-immersion objective was used, the confocal pinhole was set to three Airy resulting in quasi-wield field images. Image resolution was adjusted to a pixel size equal to 70 nm and a four lines average was achieved.

**Image analysis.** The measure of FRET between membrane and cytosolic proteins is prone to error mainly due to the high proportion of pixels with either donor alone or acceptor alone. Indeed, for FRET to happen, the two proteins must be at least expressed at a sufficient concentration in the same voxel. Although PIE-FRET is the most efficient method to select these pixels, a post-processing analysis can also be performed when acquiring with classical confocal or wield field microscopes. 3sFRET images were preprocessed by a home-made macro in ImageJ environment. Briefly, an automatic MaxEntropy filter

was applied on the Donor image while an automatic Moments filter was applied on the Acceptor image - generating a Donor and an Acceptor binary masks, respectively. These two masks were multiplicated to obtain a Donor–Acceptor mask. This latter enables the selection of pixels with significant levels of both Donor and Acceptor fluorescences. The Donor–Acceptor mask was then used to filter the three raw channels (Donor, FRET and Acceptor) before to save three independent TIFF images. Afterwards, these TIFF images were processed with PixFRET plugin in ImageJ environment[64]. We first analyzed the donor-alone images, we extracted the variables of the exponential fit of donor spectral bleed-through as a function of fluorescence intensity. We finally averaged these variables to generate to donor spectral bleed-through correction factor. In a second time, we proceed in the same way for the acceptor-alone images and extracted a constant value for the acceptor spectral bleed-through. Finally, we analyzed the three channels of FRET images and we sorted out the FRET efficiency maps. Pixel distribution histograms of these FRET efficiency maps were extracted and fitted with a robust Gaussian regression (Prism, GraphPad Software) prior their averaging. The mean value of each Gaussian regression was averaged for each condition.

## Proximity ligation assay (PLA)

We used Duolink in situ PLA mouse/rabbit (Sigma, DUO92101) to evaluate two proteins interaction (proximity <40 nm), using previously described protocol[32]. In brief, adult cardiomyocytes were fixed with 4% formaldehyde for 10 min, then permeabilized in 0.1% Triton X-100 for 15 min, and blocking was carried out for 30 min at 37 °C. Subsequently, primary antibodies that recognize the two proteins of interest were incubated overnight at 4 °C [rabbit anti-SERCA2 (Cell Signaling, 4388; 1/200) and mouse anti-GSK3β (Abcam, ab-93926; 1/200)]. Ligation of PLA probes for 30 min, amplification and detection were performed according to manufacturer's instructions. Detection was carried out by confocal microscopy[65]. A maximum intensity projection created an output image from Z stack performed at 0.5 μm depth. A total of 6–11 cells were counted in each experiment (n = 3) for quantification. We used ImageJ software (National Institutes of Health) to quantify the dots corresponding to the positive interaction of the two targeted proteins of interest.

## Animals

All animal experiments were conducted in accordance with the Claude Bernard University of Lyon ethics committee CE2A-55 (approval no.: APAFIS#19896-201903212127912v2). All experiments were performed on a parity of male and female mice of 8-12 weeks of age. Animals were housed in stable groups of four in individually ventilated cages (Nextgen - Allentown, USA – conventional animal facility) with standard nesting materials (cotton, tunnel) and ad libitum access to filtered water and standard diet (2018 global rodent diet, Envigo, France). Room temperature (housing and experiment) were maintained at 22 ± 2 °C and light cycle were at 12:12.

**Cardiomyocyte-specific Serca2 null mice.** The *Serca2*flox/flox Tg(αMHC-MerCreMer) and *Serca2*flox/flox transgenic mice were kindly provided by Experimental Medical Research at Oslo University Hospital Ulleväl (OUH-U). Animal husbandry, breeding and induction of cardiac *Serca2* gene excision were carried out using previously described protocols[36]. The primers used for genotyping mice were, for Serca2flox/flox (OL086/for: 5′.TCTTCATAACACACGCCAATTT.3′ and OL087/rev: 5′.CCCTTTGCTGCCAATTAACTATT.3′), for MerCreMer transgene (OL213/for: 5′.GCATTACCGGTCGATGCAACGAGTGATG AG.3′ and OL214/rev: 5′.GAGTGAAACGAACCTGGTCGAAATCAGTG CG.3′), and to test the *Serca2* excision (loxP1-2/for: 5′.CTTATCGAT ACCGTCGATCGGACCTC.3′ and loxP1-2/rev: 5′.TGTTTTCATAGAG GCCATGACCACAG.3′).

**Viral vectors for in vivo gene transfer.** The adeno-associated virus serotype 9 (AAV9) used to overexpress SERCA2A in our study, pAAV[Exp]-CMV-hATP2A2[NM_001681.3] (#VB200110-1239tqc), pAAV[Exp]-CMV-hSERCA2a-S663A (#VB200110-1240mkt) and pAAV[Exp]-CMV-hSERCA2a-S663E (#VB200110-1241vgn), were constructed and packaged by VectorBuilder (Santa Clara, CA). The vectors ID can be used to retrieve detailed information about the vectors on vectorbuilder.com.

**Cardiac Serca2 excision and rescue.** Mice were randomized into three groups of *Serca2a* rescue, WT, [S663A] and [S663E], to receive a jugular injection of $1 \times 10^{11}$ PFU of rescue AAV9 at 28 days before ischemia/reperfusion (I/R) surgery. A total of three intraperitoneal injection of 1 mg tamoxifen each at 18, 16 and 14 days before I/R surgery were carried out for cardiac *Serca2* knock-out induction.

**In vivo model of acute myocardial ischemia/reperfusion (I/R) injury.** Mice underwent I/R surgery with 60 min of ischemia followed by 24 h of reperfusion[32]. Briefly, mice were anesthetized with sevoflurane 4%. After an intraperitoneal injection of buprenorphine (0.1 mg/kg), the animals were orally intubated, ventilated and body temperature was maintained at 37 °C. A left thoracotomy was performed in the fourth intercostal space. A small curved needle was passed around the left anterior descending coronary artery to induce I/R. Mice were subjected to 60 min regional myocardial ischemia followed reperfusion. After 24 h reperfusion, heart was harvested under deep anesthesia, for the area at risk (AR) assessement after Evans Blue injection, and necrosis was then determined by triphenyltetrazolium staining. Infarct size was quantified with SigmaSCAN Pro5.0.0. Numbers of mice used in myocardial I/R model were: (i) 18 for *Serca2a* WT rescue (13 analyzed and 5 deaths (3 surgery, 2 reperfusion)), (ii) 12 for *Serca2a*[S663A] rescue (11 analyzed and 1 death), and (iii) 18 for *Serca2a*[S663E] rescue (13 analyzed and 5 deaths at reperfusion). No analyzed animals were excluded.

**Ex vivo cardiomyocytes**

**Cardiomyocytes isolation.** Adult mice were euthanized by cervical dislocation and the mouse heart was perfused using the Langerdorff technique with an enzymatique digestion buffer. After the optimal digestion of the heart, $CaCl_2$ lifts were realized on ventricular adult cardiac myocytes (ACM) until reaching a final calcium concentration at 0.9 mM for the day of the experiment[66].

**Hypoxia-reoxygenation.** Rod-shaped calcium-tolerant mouse cardiomyocytes were subjected to a suspension-simulated hypoxia in a controlled hypoxic chamber (Eppendorf Galaxy 48), induced by $N_2$ flushing up to 1% $O_2$ for 45 min, in a Tyrode solution (140 mM of NaCl, 5 mM of KCl, 10 mM of HEPES, 1 mM of $MgCl_2$, and 1.8 mM of $CaCl_2$ at pH 7.4 at 37 °C)[67]. Reoxygenation was induced by the addition of culture medium [MEM #21575022 Gibco®, 10% fetal bovine serum (FBS), 10 mM of BDM, 100 U/ml of penicillin, 2 mM of glutamine, and 2 mM of ATP] for 2 hours. Cardiomyocytes were then collected and homogenized for imaging or proteomic analysis.

**Mitochondrial characterization.** Live ACM were labeled with 5 μM MitoSOX red reagent (Invitrogen, M36008) for detecting mitochondrial reactive oxygen species (ROS) and 20 nM MitoProbe Tetramethylrhodamine (Invitrogen, M20036) for determining mitochondrial membrane potential. Percentage of stained cells was measured using the 561 nm laser on a LSRFortessa X-20 (Becton-Dickinson) and then analyzed using FACSDiva 8.0.1 software (Becton-Dickinson). The total ACM number threshold for flow cytometry was set at 10,000 cells.

**Mitochondrial respiration.** Mitochondrial oxygen consumption rate was measured on intact cardiomyocytes (500 μg), incubated in CCB buffer (140 mM NaCl, 5 mM KCl, 10 mM HEPES, 1 mM $MgCl_2$, 2 mM $CaCl_2$, 10 mM glucose; adjusted to pH 7.4.) at 25 °C in the Oroboros oxygraph. Kinetic studies were performed by sequential addition of pyruvate (2 mM), oligomycin (2 μM), FCCP (25 μM), and rotenone/antimycine A (0.5 μM/4 μM). Results are expressed as both ATP-linked respiration (basal) and FCCP-stimulated maximal respiration. Oxygen consumption was evaluated by the Oroboros DatLab4 software and expressed as nanomoles of oxygen per minute per milligram of total protein.

**Field-stimulated cytosolic $Ca^{2+}$ imaging.** Fluo-4-AM measurements were recorded at 37 °C with a Nikon confocal microscope (A1R) with a 40× oil-immersion objective. Cardiomyocytes loaded with 1 μM Fluo-4 (Thermo Fisher, F14201) for 30 min at 37 °C, were placed in the field stimulation buffer, and conditioning pacing (1 Hz) was used to achieve a steady-state reticular $Ca^{2+}$ loading before caffeine application (10 mM). Caffeine response parameters were further analyzed with OriginPRO (OriginLab9.0.0.87)[65].

**Sarcomere shortening.** Isolated Cardiomyocytes were paced using a field stimulator at a frequency of 1 Hz, 20 V and movement of cardiomyocytes was measured and analyzed with the IonWizard software Ionoptix[68].

**Patch-clamp electrophysiology.** Current recordings were made at room temperature under voltage clamp using the whole-cell configuration of the patch-clamp technique. Command voltage and data acquisition were performed with pClamp software (Axon Instruments, Foster city, CA, USA). The holding potential (HP) was kept at −80 mV. Membrane capacitance ($C_m$) was systematically measured and was calculated by analyzing the capacitive surge produced by a small voltage step as previously described[69]. The L-type $Ca^{2+}$ current ($I_{Ca,L}$) was evoked every 10 s by 250-ms voltage steps spaced 10 mV apart and varying between −90 and 60 mV and was measured as the difference between the peak inward current and the current at the end of the pulse. Current traces were uncorrected from the leak and normalized to $C_m$. The measurement of the Na–Ca exchange current ($I_{NCX}$) was made as described by Espinosa et al.[70]. From HP, a depolarization to −50 mV was applied for 20 ms to inactivate the transient sodium current followed by a 30-ms depolarizing step to +10 mV to activate $I_{Ca,L}$. This protocol was applied at 0.1 Hz until $I_{Ca,L}$ and the slow tail inward current recorded on repolarization reached a steady-state. The Na⁺ external solution surrounding the myocyte was then replaced in few seconds with a Li⁺ solution and $I_{NCX}$ was measured 20 ms after the onset of the repolarization to −80 mV as the Li⁺-sensitive slow tail current. $I_{NCX}$ measurement was also used to indirectly estimate the activity of WT and mutant SERCA2. The amount of charges transported by the exchanger at −80 mV was then determined by the integral of the Li⁺-sensitive tail current (integrated $I_{NCX}$) between 15 and 150 ms after the start of repolarization. For $I_{Ca,L}$ recording, the external solution contained (in mM): 136 TEACl, 2 $MgCl_2$, 1.8 $CaCl_2$, 5 4-aminopyridine, 10 glucose, 10 Hepes, adjusted to pH 7.4 with TEAOH and the internal solution contained (in mM): 140 CsCl, 1 $MgCl_2$, 3 MgATP, 10 EGTA, 10 Hepes, adjusted to pH 7.2 with CsOH. For $I_{NCX}$ recording, the external solution contained (in mM): 136 NaCl (or LiCl), 5 CsCl, 2 $MgCl_2$, 1.8 $CaCl_2$, 10 glucose, 5 Hepes, adjusted to pH 7.4 with NaOH (or LiOH) and the internal solution contained (in mM): 7 NaCl, 20 CsCl, 110 Cs-aspartate, 1.1 $MgCl_2$, 0.2 EGTA, 5 Hepes, adjusted to pH 7.2 with CsOH.

**Statistics**

Data processing and statistical analyses were conducted with Excel (Microsoft 2016), Prism (GraphPad5.0.0), OriginPro (OriginLab9.0.0.87) and R (R Core Team 2020) software. Mean with 95% confidence interval or Median with an interquartile range, calculated from at least three independent experiments, are shown. Two-tailed

unpaired *t* test with Welch's correction, two-tailed paired with Wilcoxon test, one- and two-way ANOVA followed by Tukey's multiple comparisons or a Mann–Whitney test were used to assess significance (ns $p \geq 0.05$, $*p < 0.05$, $**p < 0.01$, $***p < 0.001$, $****p < 0.0001$).

## Reporting summary

Further information on research design is available in the Nature Portfolio Reporting Summary linked to this article.

## Data availability

The data supporting this article and other findings are available within the manuscript, figures and supplementary data and from the corresponding authors upon request. Source data are provided with this paper.

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

## Acknowledgements

We would like to acknowledge our fundings from the French National Research Agency ANR-JCJC 16-CE17–0020-01 to L.G and OPeRa IHU research program ANR-10-IBHU-0004 to M.O, and from the Leducq Transatlantic Network of Excellence "Targeting mitochondria to treat heart disease" (MitoCardia, 16CVD04) to M.P and M.O. We are very grateful to William Edward Louch for the generous gift of the *Serca2^flox/flox* Tg(αMHC-MerCreMer) and *Serca2^flox/flox* transgenic mice, and his advice in maintaining the SERCA transgenic mice. We would like to thank the iXplora [https://sfrsantelyonest.univ-lyon1.fr/ixplora/index.php] platform (Celphedia) [https://celphedia.eu/en/] for in vivo expertise and help in analysis; the Tissue-Tumorotheque Est (CRB-HCL, HCL's biobank) for the human biological samples; the contribution of Protein

Science Facility at the SFR Biosciences (UMS3444/CNRS, US8/Inserm, ENS de Lyon, UCBL) for carrying out mass spectrometry experiments, especially Frederic Delolme and Adeline Page for their help; and ITMO Cancer AVIESAN (Alliance Nationale pour les Sciences de la Vie et de la Santé, National Alliance for Life Sciences and Health) within the framework of the cancer plan for Orbitrap mass spectrometer funding. Finally, we would like to thank Servier Medical Art for the generation of clipart figures (Figs. 5A, 6 and Supplementary Fig. 3A) (https://creativecommons.org/licenses/by/3.0/, accessed 5 February 2023).

## Author contributions

F.G. and L.G. designed the experiments, researched and analyzed data, contributed to the discussion, and wrote the manuscript. F.G., L.B., C.B., M.D., Y.G., G.B., C.C., C.C.D.S., S.D., S.C., C.P., F.S. and M.P. collected data. F.F., L.S., T.B., H.T. and MO performed the clinical approaches and the sampling of heart biopsies of control and failing patients. B.P., A.K., C.L. and L.G. performed the in vivo preclinical approaches. T.D. and V.G. performed in vitro experiments on hiPSC-CM cells. M.P., S.D., and C.C. contributed to the discussion and reviewed/re-edited the manuscript. L.G. is the guarantor of this work and has full access to all the data in the study.

## Competing interests

The authors declare no competing interests.
