## [Peer Review File · Nature Communications]

SERCA2 phosphorylation at serine 663 is a key regulator of Ca²⁺
+ homeostasis in heart diseasesREVIEWER COMMENTS

Reviewer #1 (Remarks to the Author):

Modulation of Ca²⁺ transport in the heart represents a possible strategy for ameliorating or reversing heart disease. A major regulator of Ca²⁺ transport in the sarcoendoplasmic reticulum (SR) of cardiomyocytes is SERCA, an integral membrane enzyme that transports 2 Ca²⁺ ions in exchange of 3 H₂O⁺ per ATP molecule hydrolyzed. SERCA is responsible for approximately 70% of Ca²⁺ ions re-uptake in the SR, and it is involved in the diastolic phase of the heart muscle. In the past years, Hajjar, Kranias, and co-workers have extensively used AVV therapy to overexpress SERCA in the cardiac muscle to re-establish proper contractility in response to dysfunctions due to genetic mutations or heart disease.

In this paper, Gomez and co-workers studied SERCA as a possible target to reverse the damages of acute myocardial ischemia-reperfusion injury. Using a wide array of techniques, these researchers found that phosphorylation of Ser663 is critical for pathophysiological events in the heart. This new study expands their previous conclusions about the effects of pharmacological inhibition of GSK3 β , a Ser/Thr kinase that targets SERCA. They found that GSK3 β phosphorylates SERCA at Ser663 both in human and mouse heart. The mass spectrometry data show the presence of phosphopeptides in the correspondence of S663. This site was also confirmed by in silico analysis of the possible sites targeted by the GSK3 β kinase. Using CRISP/Cas-9 genome editing, they engineered S663A and S663E (pseudo-phosphorylated) mutants into human cell lines. They found that the S663A mutant has a response similar to the inhibitor of the GSK3 β kinase, while the pseudo-phosphorylated mutant behaves similarly to the pS663 SERCA, although to a lesser extent. The authors concluded that inhibition of phosphorylation at Ser663 increases SERCA activity. Based on these results, they conducted a series of AAV experiments comparing WT, S663A, and S663E and measuring Ca²⁺ transient as well as electrophysiological parameters. Their results confirm the direct involvement of S663 phosphorylation in the excitation-contraction coupling (ECC). However, it is unclear to this reviewer why the electrophysiological recordings showed no difference between the cardiomyocytes of the three mutants analyzed. Finally, the authors showed compelling data linking SERCA phosphorylation at Ser663 with myocardial infarct size. Although this is probably the most striking result, the part is less developed. What are the causes for smaller myocardial infarct size?

Overall, the paper is exciting and reveals a previously ignored hot spot in SERCA regulation that constitutes the groundwork for developing new therapies. I believe the paper is suitable for publication in your journal after the authors address the following points:

- A) A clear explanation of the electrophysiological recording is needed. It is unclear to this reviewer why the different mutants do not manifest a difference in the electrophysiological response.
- B) The authors need to provide a biochemical explanation for the reduction in myocardial infarct size. This part is of clinical importance and needs to be expanded.
- C) In the heart, SERCA is regulated by phospholamban (ventricles) and sarcolipin (atria). What are the implications of S663 phosphorylation to the regulation by these small membrane proteins? In particular, phospholamban is phosphorylated by PKA and CamKII. Are these events concomitant with SERCA phosphorylation? Or are they independent?
- D) As stated by the authors, phospholamban and sarcolipin act as possible subunits of SERCA. Two new papers support this hypothesis (doi: 10.1126/sciadv.abi7154. and doi: 10.7554/eLife.66226).
- E) As a minor point, the authors need to update their citations in the introduction to emphasize the importance of regulating Ca²⁺ cycling in heart disease. In particular, they should cite the most recent papers that involve SERCA AVV by Kranias, Hajjar, and Chien laboratories (e.g., doi: 10.1093/cvr/cvac021, doi: 10.1038/s41467-022-29703-9 etc.)

Reviewer #2 (Remarks to the Author):

The paper submitted represents a state of the art approach on identifying novel mechanisms associated with I/R reperfusion on which the group has an outstanding track record going from basic science all the way to RCT.

Here they focus on potential roles of GSK3b on SERCA. The focus on the 663 P site in SERCA2. Mutatant have been produced creating a silenced or constitutive active protein for the 663 locus.

The work is highly original but needs a context. previous work has been performed using cyclosporin and post conditioning that went all the way to the clinic. So it will be essential to highlight why this novel mechanism is additive or more relevant and better then previous interventions after myocardial (or cerebral) ischemia. The group has data on SERCA phosphorylation in preclinical models (pg8) but the data for 663 are not presented.

The paper is well written and the data are presente in an orderly fashion. Crucial data is missing on the efficacy of the AAV gene transfer and more important the functional analysis of hte mouse I/R model.

A potential risk in the approach is the wide use of techniques and models going from HEK cells to the in vivo mouse model. It would be crucial to use more human material (cardiomyocytes) rather hten mouse due to the limitations of the rodent model and modest clinical significance. More important even is that throughout th epaper there is confusion on definitions. ACS is not the same as I/R is different form HR in vitro and is remote from heartfailure. these terms are used in the paper as similar or even comparable, which is not acceptable (cannot be a surprise to a recognized research team).

early data in a preclinical model that the owrk can be reproduced and is relevant for the clinic would be recommended

Details

Page 3

These results demonstrate for the first time that the S663 phosphorylation of SERCA2 is a physiological and clinical event of cardiac function, the increase of which appears to be concomitant with heart diseases such as acute myocardial infarction.

Mixing of models and clinical conditions.

Page 8

from both preclinical animal model and failing patient.

Please present the data on the 663 position from preclinical models

Page 9

We believe that our study paves the way for prospective clinical trials to evaluate the potential cardiovascular strategies to prevent phosphorylation of SERCA2 at serine 663.

Please enlighten us on the potential clinical approach.

Fig iE and F

explain the huge variation in the control setting

Reviewer #3 (Remarks to the Author):

In their article, Gonnot et al have examined direct phosphorylation of SERCA2 by GSK3B. Their work shows that phosphorylation at s663 inhibits SERCA function, and that inhibition of this mechanism is protective in ischemia and reperfusion by encouraging calcium recycling into the ER/SR. The data are novel and interesting. However, I have several criticisms. Most importantly, I have concerns about the efficacy of SERCA knockdown and replacement in the cardiomyocyte experiments, and the role of SERCA isoforms other than SERCA2a.

1. In general, there are large amounts of data in the supplement, most of which could be included in the main manuscript figures. Please refer to other articles in high profile journals to see how this is usually done. Figures with MANY panels are now the norm. With the article as presently written, it is difficult for the reader to understand the story, as it requires so much flipping to the supplement and then the supplemental figure legends.

Furthermore, the data in the supplement are not adequately referred to in the main text. For example, for supplemental Fig. 2 the authors state that SERCA2 and GSK3B are closely localized, "These results were reinforced by three different proximity and interactions analyses". The authors do not state which analyses were done, and not even which cell types were used. Strangely, the authors neglect to mention that these data show that the interaction between the two partners increases during ischemia. This seems pretty central to the story. I have similar concerns with the presentation of data in the other supplemental figures. Please fully describe what experiments you have performed and the results you've found in the main text.

2. After having identified 11 consensus phosphorylation sites by GSK3B, the authors next examined ser663 specifically using mass spec in mice and human tissue, and showed increases in diseased hearts. The leap between these two findings is a bit unusual. Did the authors check for similar alterations in the other 10 sites? Why was this site chosen for further investigation?

3. The data that seem the most problematic are those the authors present data from cardiomyocyte SERCA2 "null mice" with AAV-mediated overexpression of variants of r2663. However, this "null" label is inappropriate since previous work by the creators of this mouse line has shown that SERCA remains present at low, but functional levels until late time points after knockout. The authors state that tamoxifen injection was induced in the adult animals 18 days prior to I/R surgery. The previous work on these animals has shown that there should be considerable endogenous SERCA expression remaining at this time point, as there was in fact still demonstrable SERCA expression at 7 weeks (approx. 50 days) after excision. The data presented in Suppl Fig. 6A only show blots from AAV-treated animals, so it was not verified that endogenous SERCA levels were effectively reduced and replaced. In general, it is not straight forward to induce AAV-mediated protein expression in the adult heart, and I am surprised if jugular injection of virus would accomplish this. The presented data (which have undefined labels on the PCR and Western blots, are these animal numbers?) show that in the AAV treated individuals, total SERCA levels which are at least as high as in normal C57BL/6 animals. However, only 3 individuals of AAV animals are shown and they are compared with a single C57 mouse? Clearly a larger sample size is needed to present mean data here.

4. This reviewer's suspicion that incomplete SERCA knockout and replacement have occurred is supported by the data shown in Figure 4 and Supplemental Fig 6. At all points previous in the manuscript rS663E exhibited super-inhibitory effects on Ca handling (ie. beyond WT, and opposite changes from S663A). This does not occur in Figure 4, as values in rS663E are similar to WT. This suggests that endogenous SERCA has not be adequately replaced.

5. Another issue with these data presented in Supplemental Fig 6 is the presentation of caffeine-elicited a time to peak measurements. This value is not of interest, as it merely reflects the speed with which caffeine contacts the cell. It is the magnitude of the caffeine-elicited release which is important.

6. The mouse embryonic fibroblast cell line was also derived from the SERCA2 KO mice. However, it is unclear as presented whether this is also an inducible SERCA KO model, or whether protein levels are absent. Please provide additional information, and verify that SERCA is absent in these cells. Here it is unclear why the phosphomimetic cells (rS663E) did not exhibit higher degrees of cell death than WT during hypoxia (Suppl Fig. 3A). This is unexpected since these cells show higher cytosolic Ca levels Panels B-C).

7. The issue of SERCA isoforms does not appear to have been adequately addressed in the paper. Are SERCA1 and 3 phosphorylated in a similar manner to SERCA2? In several experiments SERCA2a has been altered, but does SERCA2b remain in an endogenous form? What role do alternative SERCA isoforms play in experiments, where only SERCA2 has been altered?

Response to Reviewers Comments

We thank the reviewers for their careful consideration of this manuscript and the positive comments on the importance of the topic, quality of our experiments, and strength of our conclusions. Based on their comments, we have thoroughly revised the manuscript by including additional experimental data and analysis to further assess the functional consequences of our findings both in our preclinical model and in human cardiomyocytes in order to support the clinical significance of our study.

We hope that the reviewers will find the revised version of our manuscript considerably improved and acceptable for publication in Nature Communications.

Reviewer #1 (Remarks to the Author):

Modulation of Ca²⁺ transport in the heart represents a possible strategy for ameliorating or reversing heart disease. A major regulator of Ca²⁺ transport in the sarcoendoplasmic reticulum (SR) of cardiomyocytes is SERCA, an integral membrane enzyme that transports 2 Ca²⁺ ions in exchange of 3 H₃O⁺ per ATP molecule hydrolyzed. SERCA is responsible for approximately 70% of Ca²⁺ ions re-uptake in the SR, and it is involved in the diastolic phase of the heart muscle. In the past years, Hajjar, Kranias, and co-workers have extensively used AVV therapy to overexpress SERCA in the cardiac muscle to re-establish proper contractility in response to dysfunctions due to genetic mutations or heart disease.

In this paper, Gomez and co-workers studied SERCA as a possible target to reverse the damages of acute myocardial ischemia-reperfusion injury. Using a wide array of techniques, these researchers found that phosphorylation of Ser663 is critical for pathophysiological events in the heart. This new study expands their previous conclusions about the effects of pharmacological inhibition of GSK3 β , a Ser/Thr kinase that targets SERCA. They found that GSK3 β phosphorylates SERCA at Ser663 both in human and mouse heart. The mass spectrometry data show the presence of phosphopeptides in the correspondence of S663. This site was also confirmed by in silico analysis of the possible sites targeted by the GSK3 β kinase. Using CRISP/Cas-9 genome editing, they engineered S663A and S663E (pseudo-phosphorylated) mutants into human cell lines. They found that the S663A mutant has a response similar to the inhibitor of the GSK3 β kinase, while the pseudo-phosphorylated mutant behaves similarly to the pS663 SERCA, although to a lesser extent. The authors concluded that inhibition of phosphorylation at Ser663 increases SERCA activity. Based on these results, they conducted a series of AAV experiments comparing WT, S663A, and S663E and measuring Ca²⁺ transient as well as electrophysiological parameters. Their results confirm the direct involvement of S663 phosphorylation in the excitation-contraction coupling (ECC). However, it is unclear to this reviewer why the electrophysiological recordings showed no difference between the cardiomyocytes of the three mutants analyzed. Finally, the authors showed compelling data linking SERCA phosphorylation at Ser663 with myocardial infarct size. Although this is probably the most striking result, the part is less developed. What are the causes for smaller myocardial infarct size? Overall, the paper is exciting and reveals a previously ignored hot spot in SERCA regulation that constitutes the groundwork for developing new therapies. I believe the paper is suitable for publication in your journal after the authors address the following points:

A) A clear explanation of the electrophysiological recording is needed. It is unclear to this reviewer why the different mutants do not manifest a difference in the electrophysiological response.

Our electrophysiological experiments had 2 main objectives: (i) to characterize the two major Ca^{2+} fluxes across the sarcolemma involved in ECC: the L-type Ca^{2+} current ($I_{\text{Ca,L}}$) and the $\text{Na}^+/\text{Ca}^{2+}$ exchange current (I_{NCX}); (ii) to use the measurements of NCX currents, in a Ca^{2+} overload context, to test the competition between sarcolemma Ca^{2+} efflux (primarily through the NCX) and Ca^{2+} uptake into SR/ER (through SERCA2).

(i) Indeed, we reported no difference neither in $I_{\text{Ca,L}}$ densities, and more strikingly nor in I_{NCX} densities between the cardiomyocytes of the three mutants analyzed.

We now presented similar recordings obtained from non-transgenic control mouse cardiomyocytes in the revised Fig. 4E-I. We show that their $I_{\text{Ca,L}}$ density were comparable to those of mutant cells (revised Fig. 4E). In contrast, their I_{NCX} density was higher than all 3 mutants (revised Fig. 4F-G). These results are in accordance with those from the literature (Sarcoplasmic reticulum function in murine ventricular myocytes overexpressing SR CaATPase, DOI: [10.1006/jmcc.1998.0834](https://doi.org/10.1006/jmcc.1998.0834) / Regulation of sarcoplasmic reticulum Ca^{2+} ATPase pump expression and its relevance to cardiac muscle physiology and pathology DOI: [10.1093/cvr/cvm056](https://doi.org/10.1093/cvr/cvm056)) and argue for a SERCA overexpression in our 3 mutants.

This limitation of our rescued model is now clearly recognized in the revised Discussion: “ The fact that overexpressing SERCA2 pumps leads to better SR Ca^{2+} transport without changing these sarcolemma electrophysiological properties has been already reported and we admittedly recognize that is the limitation of our rescued model”.

We additionally demonstrated that there is no difference in NCX protein expression level (revised suppl Fig. 3G), reinforcing the idea that the reduced INCX currents in mutants have to be related to an increased SERCA2 activity.

(ii) In the next protocol, we challenged our mutant cells by the presence of external Li^+ , which caused the cessation of the NCX-dependent Ca^{2+} efflux across the sarcolemma, creating a cytosolic Ca^{2+} overload (visually confirmed by an alteration in cardiomyocyte contraction at the time of the experiment).

On return to external Na^+ , the slow tail inward current recorded on repolarization was increased in every cardiac cell type (revised Fig. 4I). These data showed that NCX exchangers worked properly, even in our mutant cells despite the SERCA2 overexpression, in accordance with the conserved NCX protein expression (revised suppl Fig. 3G).

Despite the limitations mentioned above, we did the two following observations: (a) the return to the slow tail inward current value before external Li^+ addition was faster in any of the 3 mutants than in non-transgenic cells (revised Fig. 4I) and (b) the integrated NCX current measured after return to external Na^+ was significantly smaller in phosphoresistant rescue as compared to rWT and phosphomimetic cardiomyocytes (revised Fig. 4J).

Respectively, these results argued for a SERCA overexpression in our mutants and a reinforced SERCA pump activity in the phosphoresistant rescued cells. This latter point was supported by the results from the caffeine protocol (Fig. 4D).

Altogether, our work proved that the use of an improved SERCA2 pump is relevant to manage episodes of Ca^{2+} overload as it occurs during myocardial infarction.

We have now clarified the description of the electrophysiological recordings, on Page 6:

Monitoring cytosolic Ca^{2+} transients under field stimulation at 1Hz (Fig. 4A) demonstrated that the Ca^{2+} transient amplitude was significantly increased in rS663A as compared to rWT

and rS663E cardiomyocytes (Fig. 4B), suggesting that the SR/ER Ca^{2+} stores were larger in cardiomyocytes expressing the phosphoresistant mutant. To note, no difference in the time-to-peak was observed between mutant groups (Suppl. Fig. 3F), suggesting no change in RyR-induced Ca^{2+} release. Consecutively to electrical stimulations, we next evaluated the RyR-dependent reticular Ca^{2+} pool using caffeine stimulus (Fig. 4A, arrow). As expected, the maximal caffeine-peak amplitude of rS663E mutant was significantly reduced as compared to rS663A and rWT groups (Fig. 4C). However, we were surprised to visualize a similar answer in rS663A and rWT (Fig. 4C). This apparent discrepancy can be explained by a faster Ca^{2+} SERCA2-dependent pumping back into the SR/ER in the rS663A cardiomyocytes than in rWT and rS663E cells, confirmed by the higher rate of exponential decay in these cells (Fig. 4D). Accordingly, rS663A cardiomyocytes displayed a lower cytosolic Ca^{2+} level at baseline (Fig. 4E), supporting an enhanced SR Ca^{2+} pumping.

We next performed electrophysiological recordings in mouse cardiomyocytes from the three mutants in order to characterize the two major Ca^{2+} fluxes across the sarcolemma involved in ECC: (i) the L-type Ca^{2+} current ($I_{\text{Ca,L}}$) and (ii) the $\text{Na}^+/\text{Ca}^{2+}$ exchange current (I_{NCX}). No significant difference was reported between the capacitances and $I_{\text{Ca,L}}$ densities of rWT, rS663A, and rS663E mutant cardiomyocytes (Fig 4F). Likewise, I_{NCX} densities measured as the Li^+ -sensitive slow tail current 20 ms after the onset of repolarization at -80 mV (Fig 4G) or as the integrated form of this inward current (Fig 4H) were similar in the 3 subgroups. To note, when compared to cardiomyocytes isolated from non-infected control (Ctr) mice, their capacitance and $I_{\text{Ca,L}}$ density were comparable to those of mutant cells (Fig 4F). However, the I_{NCX} density of the Ctr cardiomyocytes was higher than all 3 mutants (Fig 4G-H), without being related to a difference in NCX protein expression (Suppl. Fig. 3G).

In the next protocol (Fig 4I-K), NCX currents, reflecting the Ca^{2+} cytosolic content, were used to test the competition between sarcolemmal Ca^{2+} efflux (primarily through the NCX) and Ca^{2+} uptake into SR/ER (through SERCA2) that governs relaxation³⁷. Cardiomyocytes were thus challenged with an episode of intracellular Ca^{2+} overload achieved by triggering the Ca^{2+} -induced Ca^{2+} release (CICR) after lithium blockade of NCX currents. In the presence of external Na^+ , each pacing protocol (Fig 4J, P₁ to P₃) triggered a brief tonic cell contraction. Cardiomyocytes gradually lost their ability to contract normally in the presence of external Li^+ (Fig 4J, P₄ to P₆) because of the cytosolic Ca^{2+} increase caused by the cessation of the NCX-dependent Ca^{2+} efflux across the sarcolemma. On return to external Na^+ , the slow tail inward current recorded on repolarization was almost tripled in every cardiac cell type (Fig. 4J: P₇ trace on return to external Na^+ versus P₃ trace before external Li^+), showing that NCX exchangers worked properly. However, the return to its value before external Li^+ addition was faster in any of the 3 mutants than in non-infected cells, and even more markedly in the rS663A mutant cardiomyocytes. When only considering the mutants, the integrated NCX current measured after return to external Na^+ was significantly smaller in the phosphoresistant rescue as compared to rWT (from P₈ to P₁₄) and

phosphomimetic (from P₇ to P₁₂) cardiomyocytes (Fig. 4K). These data suggest that the Ca²⁺ removal efficiency of the phosphoresistant SERCA2 was being so sizeable that it further reduced the NCX role in the competition for cytosolic Ca²⁺ removal, corroborating results from Fig. 4D.

Adding that basal cytosolic Ca²⁺ concentration was significantly reduced in phosphoresistant cardiomyocytes (Fig 4E), these data extended previous HEK data (Fig. 2) to cardiomyocytes showing that SERCA2 activity is controlled by its phosphorylation state on serine 663 residue in adult cardiac cells and that the phosphoresistant rS663A mutant augments SR/ER Ca²⁺ content and reduces cytosolic Ca²⁺ content.

As to the absence of difference between the three mutants, indeed, we observed no difference in the L-type Ca²⁺ current between the three rescues and compared to non-infected control cardiomyocytes (Fig. 4F). However, the integrated NCX current is effectively strongly reduced in the three different SERCA-rescued cells (Fig4H), compared to control myocytes, due to the overexpression of SERCA2 for each rescue (Suppl Fig S3D & S4B).

B) The authors need to provide a biochemical explanation for the reduction in myocardial infarct size. This part is of clinical importance and needs to be expanded.

We appreciate the reviewer's suggestion and we have now added new experiments on mouse cardiomyocytes isolated from the area at risk after an *in vivo* ischemia-reperfusion (60min-24h) which support the protective effect afforded by the S663A rescue via the cytosolic Ca²⁺ detoxification by the enhanced SERCA2 activity, resulting in improved sarcomere contraction.

Page 8: To determine the underlying mechanisms of the protection afforded by the S663A rescue, we analyzed the effect on known cardioprotective signaling pathways: no change was observed in the phosphorylation level of ERK and STAT3 after IR between the three groups (Fig. 5F,G). Interestingly, while no difference was measured at baseline in the level of phosphorylation of PLN between the three groups (Suppl. Fig. 3F), following an *in vivo* ischemia-reperfusion, we measured a significantly decreased phosphorylation of PLN in the r663E cardiomyocytes compared to either rWT or rS663A cells (Fig. 5H), with no significant effect on the interaction between SERCA and PLN (Fig. 5I). We next evaluated the functions of the mouse cardiomyocytes isolated from the area at risk after the *in vivo* ischemia-reperfusion insult. A reduced mortality with preserved mitochondrial membrane potential and diminished resting cytosolic Ca²⁺ level was measured in the S663A-rescued cardiomyocytes compared to the WT-rescue (Fig. 5 -L). Analysis of sarcomere shortening parameters further revealed an enhanced sarcomere contraction in the rS663A cells versus the rWT ones (Fig. 5M-P). Altogether, these results support that the protection afforded by the S663A relies on the increased activity of SERCA2 and the ensuing prevention of cytosolic Ca²⁺ overload, resulting in improved contraction.

C) In the heart, SERCA is regulated by phospholamban (ventricles) and sarcolipin (atria). What are the implications of S663 phosphorylation to the regulation by these small membrane proteins? In particular, phospholamban is phosphorylated by PKA and CamKII. Are these events concomitant with SERCA phosphorylation? Or are they independent?

On baseline conditions, we did not observe any difference in the level of phosphorylation of PLN between the three groups (Supp Fig3F). However, following in vivo ischemia-reperfusion, we measured a significantly decreased phosphorylation of PLN in the r663E cardiomyocytes compared to either rWT or rS663A cells (Fig5H), with no significant effect on the interaction between SERCA and PLN (Fig5I). This point is mentioned in the results section (page 8) and discussed (page9).

Since our study focuses on ventricular function, we did not study the atria and the sarcolipin level.

D) As stated by the authors, phospholamban and sarcolipin act as possible subunits of SERCA. Two new papers support this hypothesis (doi: 10.1126/sciadv.abi7154. and doi: 10.7554/eLife.66226).

This notion was mentioned in page 10:

Interestingly, we observed a decreased PLN phosphorylation in the phosphomimetic SERCA2-rescued cardiomyocytes only after I/R, which could therefore further increase the SERCA2 activity and aggravate the cytosolic Ca²⁺ overload. Emerging studies are trying to decipher the molecular interplay of the modulation of SERCA2 by its possible subunits, PLN and sarcolipin^{49, 50}. SERCA2 phosphorylation on serine 663 may be part of this process and will deserve future research.

E) As a minor point, the authors need to update their citations in the introduction to emphasize the importance of regulating Ca²⁺ cycling in heart disease. In particular, they should cite the most recent papers that involve SERCA AVV by Kranias, Hajjar, and Chien laboratories (e.g., doi: 10.1093/cvr/cvac021, doi: 10.1038/s41467-022-29703-9 etc.)

We appreciate the reviewer's suggestion and we have now focused the introduction on the role of SERCA in heart disease and its challenge.

Editorial Note: Parts of panel a of the figure on this page were drawn by using pictures from Servier Medical Art. Servier Medical Art by Servier is licensed under a Creative Commons Attribution 3.0 Unported License (<https://creativecommons.org/licenses/by/3.0/>)

Page 2 : Recent therapies have been proposed by targeting PLN modulation of SERCA2 in arrhythmia and HF^{15, 16}.

Reviewer #2 (Remarks to the Author):

The paper submitted represents a state of the art approach on identifying novel mechanisms associated with I/R reperfusion on which the group has an outstanding track record going from basic science all the way to RCT.

Here they focus on potential roles of GSK3b on SERCA. The focus on the 663 P site in SERCA2. Mutant have been produced creating a silenced or constitutive active protein for the 663 locus.

The work is highly original but needs a context. previous work has been performed using cyclosporin and post conditioning that went all the way to the clinic. So it will be essential to highlight why this novel mechanism is additive or more relevant and better then previous interventions after myocardial (or cerebral) ischemia.

The group has data on SERCA phosphorylation in preclinical models (pg8) but the data for 663 are not presented.

We understand the reviewer's point and these data have now been mentioned in the results section.

Page 10: In addition to the structural predisposition of SERCA2 to be phosphorylated at serine 663 (Suppl. Fig 1B), our mass spectrometry approach revealed also a significant increase in SERCA2 phosphorylation at serine 378 in ischemic mouse hearts.

The paper is well written and the data are presented in an orderly fashion. Crucial data is missing on the efficacy of the AAV gene transfer and more important the functional analysis of the mouse I/R model.

This point has now been better addressed in the results section.

Page 6: All analyzed mice presented a cardiomyocyte-specific excision of the endogenous *Serca2* (Suppl. Fig. 3B, averaging 84%) and a SERCA2 mutant rescue, as displayed by immunoblotting (Suppl. Fig. 3D). Validation of the AAV9 expression in the whole heart after jugular injection was further confirmed using the fluorescent AAV9-D4ER and the AAV9-LacZ (Suppl. Fig.3C).

the functional analysis of the mouse I/R model?

We did not intend to assess left ventricular function by echocardiography at 24 hours after reperfusion.

Indeed, our aim was to determine the effect of the phosphorylation level of SERCA2 at S663 on

myocardial infarct size. In this setting TTC is the gold standard technique. In addition, 24 hours after surgery, echocardiography is challenging on mice because of the suture and re-perfused myocardium is still stunned which may preclude standard measurements (like left ventricular ejection fraction) to detect a cardioprotective effect <https://www.ahajournals.org/doi/epub/10.1161/CIRCIMAGING.110.962282>. We agree with the reviewer that echocardiography will be of great interest in a future study focused on demonstrating the potential value of S663 in post ischemic heart failure development (next step of the project).

Nevertheless, we have performed additional functional experiments to decipher the underlying protective mechanisms.

Page 8: To determine the underlying mechanisms of the protection afforded by the S663A rescue, we analyzed the effect on known cardioprotective signaling pathways: no change was observed in the phosphorylation level of ERK and STAT3 after IR between the three groups (Fig. 5F,G). Interestingly, while no difference was measured at baseline in the level of phosphorylation of PLN between the three groups (Suppl. Fig. 3F), following an *in vivo* ischemia-reperfusion, we measured a significantly decreased phosphorylation of PLN in the r663E cardiomyocytes compared to either rWT or rS663A cells (Fig. 5H), with no significant effect on the interaction between SERCA and PLN (Fig. 5I). We next evaluated the functions of the mouse cardiomyocytes isolated from the area at risk after the *in vivo* ischemia-reperfusion insult. A reduced mortality with preserved mitochondrial membrane potential and diminished resting cytosolic Ca²⁺ level was measured in the S663A-rescued cardiomyocytes compared to the WT-rescue (Fig. 5 -L). Analysis of sarcomere shortening parameters further revealed an enhanced sarcomere contraction in the rS663A cells versus the rWT ones (Fig. 5M-P). Altogether, these results support that the protection afforded by the S663A relies on the increased activity of SERCA2 and the ensuing prevention of cytosolic Ca²⁺ overload, resulting in improved contraction.

A potential risk in the approach is the wide use of techniques and models going from HEK cells to the in vivo mouse model. It would be crucial to use more human material (cardiomyocytes) rather than mouse due to the limitations of the rodent model and modest clinical significance.

We agree with the reviewer's comment and we have now performed additional experiments on human induced pluripotent stem cell-derived cardiomyocytes to validate the effect of SERCA2 phosphorylation on Ser663 in human cardiomyocytes.

Page 5: To validate the ubiquitous role of SERCA2 phosphorylation on S663, we performed a similar experiment in human induced pluripotent stem cell-derived cardiomyocytes (hiPSC-CM) since SERCA2 plays a crucial role in cardiomyocytes³⁵. hiPSC-CM were infected by either a WT SERCA2, a phosphoresistant SERCA2 mutant (SERCA2[S663A]) or a phosphomimetic SERCA2 mutant (SERCA2[S663E]) (Fig. 2F). Similarly to what was observed in the HEK cells, the SERCA2[S663A] hiPSC-CM displayed significantly increased ER Ca²⁺ refilling rate and ER Ca²⁺ content versus the SERCA2[WT] cells (Fig. 2G-H). On the other hand, the SERCA2[S663E] hiPSC-CM showed no difference in their ER Ca²⁺ refilling rate compared to the SERCA2[WT] cells, but a significant decrease in the ER Ca²⁺ content compared to both SERCA2[WT] and SERCA2[S663A] hiPSC-CM (Fig. 2G-H).

Taken together, these results demonstrate that specific inhibition of SERCA2 phosphorylation at S663 residue increases SERCA2 pump activity in several cell types, notably hiPSC-CM.

More important even is that throughout the paper there is confusion on definitions. ACS is not the same as I/R is different from HR in vitro and is remote from heartfailure. these terms are used in the paper as similar or even comparable, which is not acceptable (cannot be a surprise to a recognized research team).

We apologize for the lack of clarity. The different notions have been clarified all along the manuscript.

early data in a preclinical model that the work can be reproduced and is relevant for the clinic would be recommended

This point has now been addressed with the hiPSC-CM experiments described above.

Details

Page 3

These results demonstrate for the first time that the S663 phosphorylation of SERCA2 is a physiological and clinical event of cardiac function, the increase of which appears to be concomitant with heart diseases such as acute myocardial infarction. Mixing of models and clinical conditions.

We apologize for the confusion and have corrected the sentence, page3:

These results demonstrate for the first time that the S663 phosphorylation of SERCA2 is increased by ischemia-reperfusion in a preclinical model, an increase also present in the human failing heart.

Page 8

from both preclinical animal model and failing patient.

Please present the data on the 663 position from preclinical models

This point has been addressed above in the first comment of the reviewer.

Page 9

We believe that our study paves the way for prospective clinical trials to evaluate the potential cardiovascular strategies to prevent phosphorylation of SERCA2 at serine 663. Please enlighten us on the potential clinical approach.

This point has been mentioned in Page 11:

We believe that our study paves the way for prospective clinical trials to evaluate the potential cardiovascular strategies to prevent phosphorylation of SERCA2 at serine 663, notably by immunotherapy approach.

Fig iE and F

explain the huge variation in the control setting

These variations represent the interindividual variation.

Reviewer #3 (Remarks to the Author):

In their article, Gonnot et al have examined direct phosphorylation of SERCA2 by GSK3B. Their work shows that phosphorylation at s663 inhibits SERCA function, and that inhibition of this mechanism is protective in ischemia and reperfusion by encouraging calcium recycling into the ER/SR. The data are novel and interesting. However, I have several criticisms. Most importantly, I have concerns about the efficacy of SERCA knockdown and replacement in the cardiomyocyte experiments, and the role of SERCA isoforms other than SERCA2a.

1. In general, there are large amounts of data in the supplement, most of which could be included in the main manuscript figures. Please refer to other articles in high profile journals to see how this is usually done. Figures with MANY panels are now the norm. With the article as presently written, it is difficult for the reader to understand the story, as it requires so much flipping to the supplement and then the supplemental figure legends.

We appreciate the reviewer's suggestion to combine the supplemental figures with the main ones: this has now been done.

Furthermore, the data in the supplement are not adequately referred to in the main text. For example, for supplemental Fig. 2 the authors state that SERCA2 and GSK3B are closely localized, "These results were reinforced by three different proximity and interactions analyses". The authors do not state which analyses were done, and not even which cell types were used.

Strangely, the authors neglect to mention that these data show that the interaction between the two partners increases during ischemia. This seems pretty central to the story.

We agree with the comment and we have now better described these results.

Page 4: Interestingly, S663 is positioned on the cytosolic side of SERCA2 and easily accessible to kinases (Fig. 1G), notably GSK3 β . To determine a potential involvement of GSK3 β on SERCA2 phosphorylation, we used three different strategies to assess the probability of the physical interaction between GSK3 β and SERCA2 and the effect of simulated ischemia-reperfusion. First, immunoprecipitation assays demonstrated that GSK3 β formed a complex with SERCA2 in both basal condition and after hypoxia-reoxygenation (H/R) in HEK cells (Fig. 1H). Second, our designed fluorescence resonance energy transfer (FRET) system validated the intermolecular interaction of GSK3 β with SERCA2 in single live

HEK cells (Fig. 1I; Suppl. Fig. 1B-D), with a FRET efficiency of 3.11% in normoxic condition (Fig. 1I), which exhibited a significant 3-fold increase following H/R (Fig. 2C). Third, a proximity ligation assay demonstrated an increased proximity between endogenous GSK3 β and SERCA2 proteins in isolated mouse cardiomyocytes after H/R (Fig. 1J). Accordingly, these results provide valuable insight regarding a novel regulatory mechanism of SERCA2, based on phosphoserine-mediated recruitment of GSK3 β to SERCA2 in the heart during an ischemic stress.

I have similar concerns with the presentation of data in the other supplemental figures. Please fully describe what experiments you have performed and the results you've found in the main text.

This has now been addressed.

2. After having identified 11 consensus phosphorylation sites by GSK3B, the authors next examined ser663 specifically using mass spec in mice and human tissue, and showed increases in diseased hearts. The leap between these two findings is a bit unusual. Did the authors check for similar alterations in the other 10 sites? Why was this site chosen for further investigation?

Our mass spectrometry analysis did not reveal any phosphorylation on the other 10 sites, therefore we focused our study on the ser663.

3. The data that seem the most problematic are those the authors present data from cardiomyocyte SERCA2 "null mice" with AAV-mediated overexpression of variants of r2663. However, this "null" label is inappropriate since previous work by the creators of this mouse line has shown that SERCA remains present at low, but functional levels until late time points after knockout. The authors state that tamoxifen injection was induced in the adult animals 18 days prior to I/R surgery. The previous work on these animals has shown that there should be considerable endogenous SERCA expression remaining at this time point, as there was in fact still demonstrable SERCA expression at 7 weeks (approx. 50 days) after excision.

The data presented in Suppl Fig. 6A only show blots from AAV-treated animals, so it was not verified that endogenous SERCA levels were effectively reduced and replaced. In general, it is not straight forward to induce AAV-mediated protein expression in the adult heart, and I am surprised if jugular injection of virus would accomplish this. The presented data (which have undefined labels on the PCR and Western blots, are these animal numbers?) show that in the AAV treated individuals, total SERCA levels which are at least as high as in normal C57BL/6 animals. However, only 3 individuals of AAV animals are shown and they are compared with a single C57 mouse? Clearly a larger sample size is needed to present mean data here.

We apologize for the lack of clarity. We agree that the term null may not be the most adapted and we have changed it to knock-down since around 84% of SERCA2 is removed (Supp Fig3B). For the AAV-mediated protein expression, we have now added more data from C57BL6 mice and clarify the labels (indeed different mouse number, supp Fig3D). Finally, to validate the jugular injection as previously performed by our collaborator (Paillard et al, Cell Reports 2017 10.1016/j.celrep.2017.02.032) and others (doi: [10.1038/mt.2013.289](https://doi.org/10.1038/mt.2013.289) ; [10.1038/gt.2010.105](https://doi.org/10.1038/gt.2010.105) ; [10.1016/j.ymthe.2006.03.014](https://doi.org/10.1016/j.ymthe.2006.03.014)), we have performed jugular injections with an AAV9-D4ER (fluorescent reporter) and an AAV9-LacZ. This point has now been better addressed in the results section.

Page 6: All analyzed mice presented a cardiomyocyte-specific excision of the endogenous *Serca2* (Suppl. Fig. 3B, averaging 84%) and a SERCA2 mutant rescue, as displayed by immunoblotting (Suppl.

Fig. 3D). Validation of the AAV9 expression in the whole heart after jugular injection was further confirmed using the fluorescent AAV9-D4ER and the AAV9-LacZ (Suppl. Fig.3C).

4. This reviewer's suspicion that incomplete SERCA knockout and replacement have occurred is supported by the data shown in Figure 4 and Supplemental Fig 6. At all points previous in the manuscript rS663E exhibited super-inhibitory effects on Ca handling (ie. beyond WT, and opposite changes from S663A). This does not occur in Figure 4, as values in rS663E are similar to WT. This suggests that endogenous SERCA has not been adequately replaced.

We understand the reviewer's point. Indeed, on that experiment (L-type Ca²⁺ and NCX currents), the rS663E cells reacted similarly to the WT ones. Nevertheless, the new comparison with non-infected control cardiomyocytes supports the efficient infection with either constructs (Fig4G-H).

Page 7: To note, when compared to cardiomyocytes isolated from non-infected control (Ctr) mice, their capacitance and I_{Ca,L} density were comparable to those of mutant cells (Fig 4F). However, the I_{NCX} density of the Ctr cardiomyocytes was higher than all 3 mutants (Fig 4G-H), without being related to a difference in NCX protein expression (Suppl. Fig. 3G).

5. Another issue with these data presented in Supplemental Fig 6 is the presentation of caffeine-elicited a time to peak measurements. This value is not of interest, as it merely reflects the speed with which caffeine contacts the cell. It is the magnitude of the caffeine-elicited release which is important.

The caffeine-elicited amplitude is displayed in Fig4C and described as follows, page 6:

As expected, the maximal caffeine-peak amplitude of rS663E mutant was significantly reduced as compared to rS663A and rWT groups (Fig. 4C). However, we were surprised to visualize a similar answer in rS663A and rWT (Fig. 4C). This apparent discrepancy can be explained by a faster Ca²⁺ SERCA2-dependent pumping back into the SR/ER in the rS663A cardiomyocytes than in rWT and rS663E cells, confirmed by the higher rate of exponential decay in these cells (Fig. 4D). Accordingly, rS663A cardiomyocytes displayed a lower cytosolic Ca²⁺ level at baseline (Fig. 4E), supporting an enhanced SR Ca²⁺ pumping.

6. The mouse embryonic fibroblast cell line was also derived from the SERCA2 KO mice. However, it is unclear as presented whether this is also an inducible SERCA KO model, or whether protein levels are absent. Please provide additional information, and verify that SERCA is absent in these cells. Here it is unclear why the phosphomimetic cells (rS663E) did not exhibit higher degrees of cell death than WT during hypoxia (Suppl Fig. 3A). This is unexpected since these cells show higher cytosolic Ca levels Panels B-C).

For generation of cells stably expressing a rescued SERCA2A cDNA in Serca2-null background, called SKOiS cells (for Serca2 Knock-Out and induction of SERCA2A), MEF-T cell line was co-transfected with 2 DNAs. A plasmid for CRE expression and which allows the excision of the endogenous Serca2. A linear DNA allowing optimal integration into the host genome of the hSerca2a rescue, [S663A] mutant or [S663E] mutant). To ensure the transfection of these 2 DNAs, the cells were selected with G418 and Hygromycin B, allowing to select the rescue and the CRE respectively.

The absence of increased cell death in the rS663E MEFs may rely on the cell type. Indeed, the rS663E in vivo AAV infection significantly increased infarct size (Fig5C).

7. The issue of SERCA isoforms does not appear to have been adequately addressed in the paper. Are SERCA1 and 3 phosphorylated in a similar manner to SERCA2? In several experiments SERCA2a has been altered, but does SERCA2b remain in an endogenous form? What role do alternative SERCA isoforms play in experiments, where only SERCA2 has been altered?

Unfortunately, we were not able to find good antibodies to perform these analyses. However, as shown in <https://www.ncbi.nlm.nih.gov/pmc/articles/PMC4159674/>, SERCA2A is the major isoform in ventricular myocytes. This point is now discussed Page 10.

SERCA2 is the major isoform expressed in cardiomyocytes³⁵ and thus the focus of our study.

REVIEWER COMMENTS

Reviewer #1 (Remarks to the Author):

The authors addressed all my comments and concerns. Congratulations for this excellent paper.

Reviewer #2 (Remarks to the Author):

The paper has improved by looking into human cell models.

Yet several questions from the first review are evaded, which is especially true for functional data.

The data on ERK and Stat are added which is very important to place the findings in perspective.

This is important in the mouse I/R model. Here data on myocardial function need to be provided for all 3 groups. The only real functional data provided is sarcomere shortening in adult mouse cells. Yet no data on the organ level or treated human iPSCM.

Serca is phosphorylated on various sites with impact on protein activity also 378. If considering a clinical immunotherapy based approach why focus on 663 or GSK? Still is not clear.

In addition to the PLN site 16; PLN can be phosphorylated at position 17 (CamK) especially during pathophysiology.

Many figures only data is shown comparing wt to (A) where (E) is missing?

Why the selection in presentation of the data?

Reviewer #3 (Remarks to the Author):

The authors have addressed my previous comments with alterations to the text, and inclusion of new data. I feel that this revised manuscript is much improved, and have no further comments.

ANSWERS TO REVIEWER COMMENTS

We thank the reviewers 1 & 3 for their enthusiastic response to our additional experiments.

Reviewer #1 (Remarks to the Author):

The authors addressed all my comments and concerns. Congratulations for this excellent paper.

Reviewer #2 (Remarks to the Author):

The paper has improved by looking into human cell models.

Yet several questions from the first review are evaded, which is especially true for functional data.

The data on ERK and Stat are added which is very important to place the findings in perspective.

This is important in the mouse I/R model.

We agree with the reviewer and we will discuss these data in the discussion, by stressing the notion that the cardioprotection afforded by the inhibition of SERCA2 phosphorylation does not depend on the activation of the RISK and SAFE signaling pathways, but on the detoxification of the cytosolic Ca²⁺ overload, as shown by the reduced cytosolic Ca²⁺ level in cardiomyocytes after *in vivo* I/R.

Page 9, discussion: "Interestingly, the cardioprotection afforded by the inhibition of SERCA2 phosphorylation did not depend on the activation of the RISK and SAFE signaling pathways, but on the detoxification of the cytosolic Ca²⁺ overload, as shown by the reduced cytosolic Ca²⁺ level in cardiomyocytes after *in vivo* I/R. Therefore, the cardioprotective effect of the phosphoresistant SERCA2[S663A] mutant may be mainly due to the increase of Ca²⁺ uptake into SR/ER, which consequently prevents both cytosolic and mitochondrial Ca²⁺ overload at reperfusion, limiting cell death and subsequent cardiac damages after ischemia-reperfusion while improving cardiomyocyte contractility (Fig. 6)

Here data on myocardial function need to be provided for all 3 groups. the only real functional data provided is sarcomere shortening in adult mouse cells. Yet no data on the organ level or treated human iPSCM.

Our study was focused on the role of SERCA2 phosphorylation on the regulation of Ca²⁺ signaling and cell death. As stated by the reviewer, we performed additional experiments on cardiomyocytes isolated from the ischemic myocardium to demonstrate the effect of the inhibition of SERCA2 phosphorylation on Ca²⁺ signaling (Fig 5L), cell death (Fig 5J) and cardiomyocyte contractility after the *in vivo* I/R (Fig 5M,5N,5O,5P). This study did not aim to assess the effect on long-term myocardial function, which we will consider for a future study. This limitation has now been stated in the discussion part, page 10:

Our study focused on the role of SERCA2 phosphorylation on cell death; however, one could wonder whether the decreased cell death and improved cardiomyocyte contractility afforded by SERCA2 phosphorylation inhibition could lead to improved post-ischemic myocardial function, paving the way for a future study.

Serca is phosphorylated on various sites with impact on protein activity also 378. If considering a clinical immunotherapy based approach why focus on 663 or GSK? Still is not clear.

We apologize if it was not enough clear. This has been further addressed in the discussion, mentioning that an additional study will be required to address if the phosphorylation on Serine 378 plays a role during myocardial infarction, and therefore may open the path for a multi-targeted

immunotherapy, even though phosphorylation on Ser378 was not identified in the human samples as now stated in the results, limiting its translational value.

Page 4, results: “Phosphorylation of SERCA2 on S378 and S555 was also identified in the mouse model; however, only the S378 phosphorylation was significantly increased in the ischemic mouse heart averaging $22.43\pm 0.3\%$ vs $6.65\pm 0.09\%$ in control group. These sites were not further investigated as they were not identified as *in silico* putative GSK3 β phosphorylation sites, and were not identified by the MS approach on human samples.”

Page 10-11, discussion: “In addition to the structural predisposition of SERCA2 to be phosphorylated at serine 663 (Fig 1G), our mass spectrometry approach revealed also a significant increase in SERCA2 phosphorylation at serine 378 in ischemic mouse hearts. Since this serine residue was not sorted out by the predictive analysis of phosphorylation sites by GSK3 β , we chose to focus our analysis only on the serine 663. Nevertheless, it will be interesting to investigate in-depth the role and function of this GSK3 β -independent phosphorylation site of SERCA2 during cardiovascular diseases, potentially opening the path for a multi-targeted immunotherapy, even though phosphorylation on serine 378 was not identified in the human samples, limiting its translational value.”

In addition to the PLN site 16; PLN can be phosphorylated at position 17 (CamK) especially during pathophysiology.

We thank the reviewer for this comment. Immunoblot of figure 5H was effectively performed with an anti-PhosphoS16/T17 PLN after *in vivo* I/R; while immunoblot of supplemental figure 3E was performed with an anti-PhosphoS16 PLN in baseline cardiomyocytes. We have now repeated the analysis of these last samples with the anti-PhosphoS16/T17 PLN and replaced supp Fig3E. Still no difference was observed at baseline in PLN phosphorylation. Additionally, the reference of anti-PhosphoS16/T17 PLN antibody was added to the methods.

Page 20, results: “(F-I) Evaluation of cardioprotective signaling pathways in lysates of cardiac area at risks, with quantification of western blotting for Phospho-ERK over total ERK1-2 (F), (G) Phospho-STAT3 over total STAT3 and (H) Phospho S16-T17-PLN over total PLN, expressed as fold of rWT. “

Page 22, results: (E) “Dot plot of phosphorylation of phospholamban at serine 16 and threonine 17, normalized with total phospholamban of basal isolated cardiomyocytes. Median with interquartile range is shown and one-way ANOVA followed by Tukey's multiple comparisons test was used to assess significance (ns $p\geq 0.05$).”

Page 28, methods: “mouse anti-PLN (Abcam, ab-2865; 1/4000); rabbit anti pS16-T17-PLN (Cell signaling 8496; 1/4000)”

Many figures only data is shown comparing wt to (A) where (E) is missing? Why the selection in presentation of the data?

Since we aimed to determine the functional mechanisms involved in the protection afforded by the inhibition of SERCA2 phosphorylation, we focused our new analyses only on the WT rescued-mice and the SERCA2-phosphoresistant mice. Moreover, as now reported in the results part, the phosphomimetic mutant induced a high mortality after IR (27%) versus WT (11%) and

phosphoresistant mutant (8%). Therefore, for ethical concern, additional experiments on the phosphomimetic mutant appeared deleterious and not adapted. This is now specified in the results part.

Page 8, results: Conversely, most data points for rS663E rescued mice were above the rWT regression line (Fig. 5C,E), indicating that the rS663E hearts aggravated the I/R injury as exhibited by a significantly higher infarct size, averaging 44.7% of AR (Fig. 5D). Additionally, it is worth noticing that the phosphomimetic rS663E mutant induced a higher mortality after I/R (27%) versus rWT (11%) and phosphoresistant rS663A mutant (8%).

Page 9, results: “We next evaluated the functions of the mouse cardiomyocytes isolated from the AR after the *in vivo* ischemia-reperfusion insult in a new set of WT-rescued or SERCA2-phosphoresistant mice to specifically address the cardioprotective effect afforded by the inhibition of SERCA2 phosphorylation. In front of the higher mortality induced by the phosphomimetic mutant rescue after I/R, this group was not included in the cardiomyocyte experiments for ethical concerns.”

Reviewer #3 (Remarks to the Author):

The authors have addressed my previous comments with alterations to the text, and inclusion of new data. I feel that this revised manuscript is much improved, and have no further comments.

REVIEWERS' COMMENTS

Reviewer #2 (Remarks to the Author):

This time all my questions have been addressed.
Clinical potential role remains unclear